# Seasonality and mobility: An Integrative framework for reconstructing Kura-Araxes pastoral systems at Maxta I, Nakhchivan

Gwendoline Maurer[1]*, Delphine Frémondeau[1], Narmin Ismayilova[2], Safar Ashurov[3], Veli Bakhshaliyev[4], Fidan Khalafova Aliyeva[3], Alexandra Nederbragt[5], Anne-Lise Jourdan[6], David Wengrow[1], Rachel Wood[7], Louise Martin[1], Rhiannon E. Stevens[1]

1 Institute of Archaeology, University College London, London, United Kingdom, 2 Oxford Nizami Ganjavi Centre, Faculty of Asian and Middle Eastern Studies, University of Oxford, Oxford, United Kingdom, 3 Institute of Archaeology and Anthropology, Azerbaijan National Academy of Sciences, Baku, Azerbaijan, 4 Department of Archaeology, Nakhchivan, Azerbaijan National Academy of Sciences, Nakhchivan State University, Azerbaijan, 5 School of Earth and Environmental Sciences, Cardiff University, Cardiff, United Kingdom, 6 Department of Earth Sciences, Bloomsbury Environmental Isotope Facility (BEIF), University College London, London, United Kingdom, 7 Oxford Radiocarbon Accelerator Unit, University of Oxford, Oxford, United Kingdom

☉ These authors contributed equally to this work.
* maurerg@cardiff.ac.uk

## Abstract

The Kura-Araxes Culture (3500–2500 BCE) is often depicted as a homogeneous pastoralist horizon, yet its internal economic and mobility strategies remain poorly understood. This study for the first time introduces an integrative framework for reconstructing site-specific pastoral practices through a detailed case study of Maxta I (Nakhchivan). It represents the first study in the Kura-Araxes context and more broadly in the Caucasus to combine zooarchaeology, Zooarchaeology by Mass Spectrometry (ZooMS), stable isotope analysis, and settlement data. This multi-method approach offers a comprehensive view of livestock management, seasonal movement, and socio-economic organisation. The results reveal a seasonally flexible agro-pastoral system that blends permanent settlement features with structured herd mobility, birthing seasons, and possible targeted secondary product exploitation strategies for fleece. Rather than adhering to a binary model of nomadic versus sedentary lifeways, Maxta I demonstrates how Kura-Araxes communities dynamically adapted to diverse ecological and social landscapes. This research challenges assumptions of cultural uniformity and establishes a new comparative model for understanding the diversity of pastoral strategies across Southwest Asia.

## Introduction

The Kura-Araxes cultural complex (3500–2500 BCE) is a geographically and spatially enduring archaeological phenomenon that emerged towards the end of

**Data availability statement:** All data generated in this study are available in the Main Text or supplemental information section of this paper. All necessary permits were obtained for the described study, which complied with all relevant regulations. All ZooMS spectra for identified samples are available on Mendeley at https://data.mendeley.com/datasets/24dyjf-7c5j/2 and are publicly available as of the date of publication. This paper does not report original code. Any additional information required to reanalyse the data reported in this paper is available from the lead contact upon request.

**Funding:** This research was funded by the London Arts & Humanities Partnership (LAHP) (Authors: GM), the UCL Institute of Archaeology Small Research Grant (Authors: GM), the Ministry of Education of the Republic of Azerbaijan (AUthors: NI) and the National Environmental Isotope Facility (NEIF Grant 2777) (Authors: GM & NI) https://www.lahp.ac.uk/ https://edu.gov.az/en/ https://www.isotopesuk.org/ https://www.ucl.ac.uk/archae-ology/ The sponsors or funders did not play any role in the study design, data collection and analysis, decision to publish, or preparation of the manuscript.

**Competing interests:** The authors have declared that no competing interests exist.

the Late Chalcolithic and Early Bronze Age in the South Caucasus and Eastern Anatolia, eventually spreading across a vast yet selective area of Southwest Asia. The origins of the complex remain debated, with evidence from Nakhchivan playing a central role [1–3]. By the 3rd millennium BCE, it had reached northwestern and central Iran, the Upper Euphrates Valley, the Amuq Plain, and the Southern Levant [4,5]. The complex was first defined by A. Kuftin in the late 1930s, outlining its traits, distribution, and periodisation to the early 3rd millennium BCE [6]-Currently the complex is known by various regional names: the Early Transcaucasian Culture in the Caucasus, the Karaz culture in Anatolia, the Yanik culture in Iran, and the Red-Black Burnished Ware (RBBW) [7] and Khirbet Kerak culture (KKW) [8] in the Levant.

Defined by a distinct "cultural package", including red and black burnished ceramics [9,10] and distinct pottery technology [11,12], zoomorphic figurines [13], distinct chipped and ground stone assemblages [14,15], distinct metallurgical traditions [16–18], funerary traditions [19], and portable hearths [20] – scholars have long debated whether this material assemblage represents a cohesive community or the widespread adoption of shared practices across Southwest Asia in the Early Bronze Age [5]. While its material culture suggests strong unifying traits, the adaptability of Kura-Araxes communities across diverse regions raises questions about the mechanisms of its spread – whether through migration, trade, or cultural exchange [21–24].

This debate is further complicated by the Kura-Araxes' theorised egalitarian socio-economic organisation, which contrasts sharply with the hierarchical structures of neighbouring cultures, such as the Maikop complex (3650–3000 BCE) to the north, the Early Kurgan complex (2400–2100 BCE) towards the later phases of the Kura-Araxes, and the Uruk phenomenon to the south (4000–3100 BCE) [25,26]. Understanding these dynamics is crucial for unravelling the cultural and economic interactions of early complex societies in Southwest Asia.

The degree of uniformity and homogeneity within the Kura-Araxes complex remains a key subject of debate [27]. Some scholars argue for a unified system, suggesting that the material and socio-economic traits represent the spread of populations adhering to shared traditions across Southwest Asia [28]. Others view the Kura-Araxes complex as monolithic in its material culture but diverse in its social structures [29]. Meanwhile, other scholars caution against equating material markers with distinct communities, proposing that the Kura-Araxes presence in regions such as Iran and the Levant may result from either population movements [11,30] or the local adoption and imitation of Kura-Araxes traditions [31].

The assumption of cultural homogeneity within the Kura-Araxes complex extends to its pastoral systems, particularly in interpretations of livestock management and subsistence strategies. The Kura-Araxes communities are frequently depicted as nomadic pastoralists or practitioners of transhumance, exhibiting varying degrees of mobility [32–35]. Several researchers propose that mobile pastoralism was central to the complex's spread during the Early Bronze Age [36–38]. Others, however, question this model and instead argue for a settled

agro-pastoralist economy [24,39,40]. Some [21] critique the overemphasis on pastoral mobility in Kura-Araxes research, while others [35] describe these communities as engaged in unspecialised animal husbandry rather than a single, uniform pastoral system [35,41]. Despite the frequent interchangeability of terms such as pastoralism, nomadic pastoralism, transhumance, agro-pastoralism, and unspecialised animal husbandry, no single descriptive or terminological definition has been consistently applied to the Kura-Araxes complex [42]. Current models for Kura-Araxes mobility and animal-based subsistence strategies are largely inferred from settlement patterns, ceramic traditions, and ethnographic analogies [24,32]. Only recently have zooarchaeological and archaeobotanical data been integrated into these discussions [41,43,44].

Only twenty sites across the Kura-Araxes oikumene (out of ~700 alone in the Caucasus) have undergone zooarchaeological analysis [4,16,45–49]. Isotopic research in the Caucasus has primarily analysed bone collagen from humans [50], animals [51], and archaeobotanical remains [52]. Sequential stable isotope analysis of herbivore enamel has been performed in Turkey [53], Armenia [54–59], Georgia [16,60,61], the North Caucasus [62], and Iran [63]. Within the Kura-Araxes Culture, such studies are increasing but remain comparatively few, and each new dataset contributes to refining our understanding of herding and mobility in the region.

A shift in settlement patterns from the Late Chalcolithic to the Early Bronze Age has also been cited as evidence for a high degree of mobility among Kura-Araxes communities. Small Early Bronze Age settlements with thin cultural deposits suggest short-term occupations [5,38]. Additionally, Kura-Araxes settlements are found at both low and high altitudes, often interpreted as evidence of transhumance [27,38,64]. The portable nature of Kura-Araxes material culture, such as andirons, ceramics with lids and large handles, and lightweight wattle-and-daub buildings has further supported arguments for high mobility [35,36]. However, some evidence contradicts the assumption of widespread mobility. Archaeobotanical studies from Sos Höyük (1700m) suggest a year-round settled agro-pastoral model [44]. Additionally, smaller sites (<5 ha) with short-lived occupations may reflect kinship-based households, where settlements were maintained for only one or two generations [5].

To investigate these competing models, our study aims to fill critical gaps in the understanding of Kura-Araxes mobility and economic organisation. We combine zooarchaeological methods, Zooarchaeology by Mass Spectrometry (ZooMS), stable isotope analyses and settlement data to examine livestock management, seasonality, land use, and settlement dynamics at Maxta I (Nakhchivan). This study provides the first site-specific characterisation of Kura-Araxes pastoral practices and mobility using an integrative, multi-proxy approach. By integrating these diverse analytical approaches, this study challenges traditional assumptions of cultural homogeneity and provides a new framework for site-specific analyses of Kura-Araxes pastoral systems. The results from Maxta I contribute to a comparative framework applicable across Southwest Asia, offering a localised perspective that complements evidence from other regions of the South Caucasus; including Armenia [54–59], Georgia [16,60,61], and northwestern Iran [63], to advance a more comprehensive understanding of pastoral strategies.

The Maxta I site (803 m asl, 39°35'23.61" N, 44°56'21.95" E) is located on the Sharur Plain in Nakhchivan, Azerbaijan (Fig 1). Initial research at Maxta I began in 1988 under the direction of V.H. Aliyev [65]. Excavations, which initially covered an area of 100 square meters, continued in 1989 to a depth of 4.60 meters. In 2006, a joint Azerbaijani-American expedition, led by S.H. Ashurov, L. Ristvet, and V.B. Bakhshaliev, conducted a sounding in a 2x2 meter area at Maxta I [66–68]. Between 2008 and 2015, Maxta I was the focus of long-term research carried out by an archaeological expedition of the Institute of Archaeology and Ethnography of the Azerbaijan National Academy of Sciences (ANAS), under the direction of S.H. Ashurov [69–71].

Maxta I is located at ~800 m asl on the Sharur Plain, bordered by the highlands of the Lesser Caucasus (Fig 2). The site lies within a semi-arid steppe zone characterised by hot, dry summers (30–35°C), cold winters (0 to –7°C), and peak precipitation in spring and autumn (Fig 2) [72]. The Sharur Plain is snow-free for c. 11 months per year, with average snow cover not exceeding 25 mm, indicating that local pastures remain accessible year-round

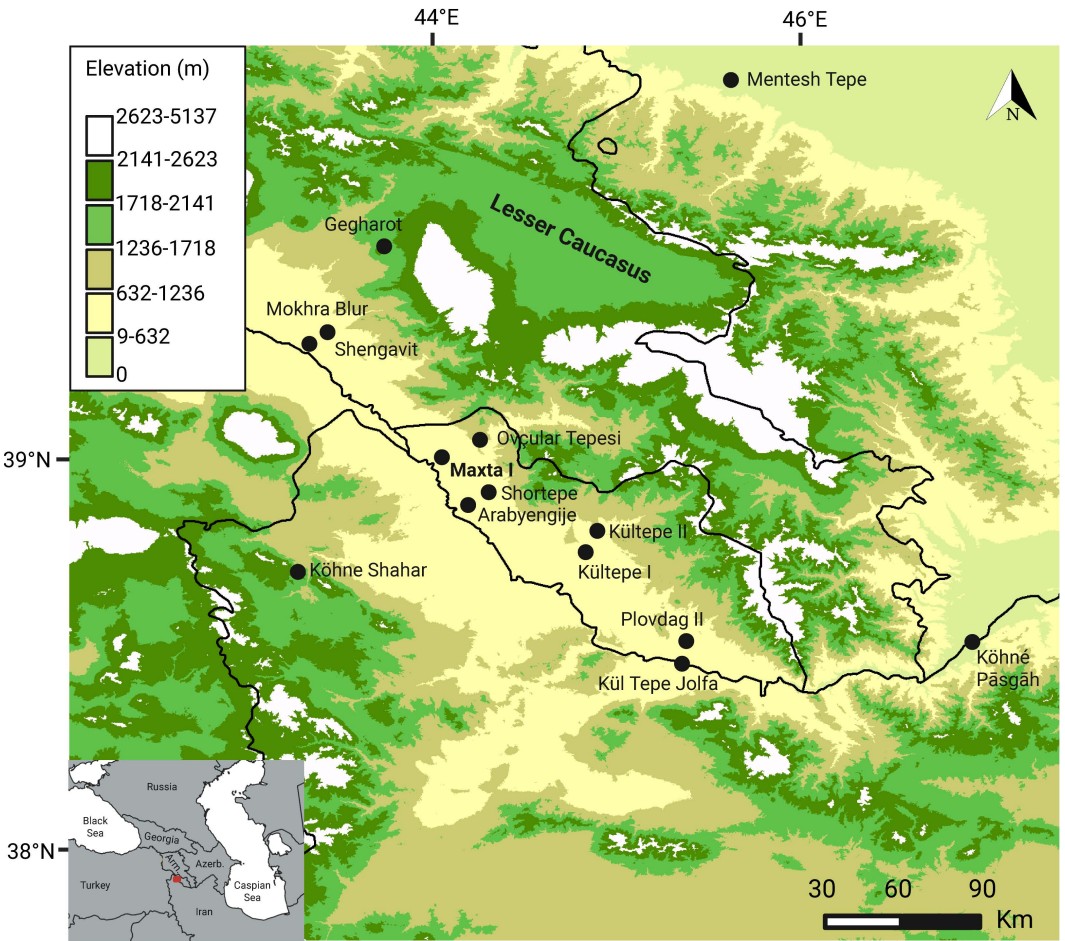

**Fig 1. Location and elevation map of Maxta I and regional Kura-Araxes sites.** Created using Aster Global Digital Elevation Model V003 from https://www.earthdata.nasa.gov/. Data manipulated using ArcMap V10.8.2.

(Fig 2). Snowfall in the surrounding highlands contributes meltwater to the Arpaçay River, which supports year-round water availability in the plain. Freshwater sources include glacial melt, rivers, aquifers, highland lakes, and natural springs [73,74].

Modern modelled $\delta^{18}O$ values in precipitation at Maxta I range from −15.4‰ in winter to −1.8‰ in summer [75], influenced by atmospheric circulation patterns such as the North Atlantic Oscillation (NAO) [76]. The regional vegetation reflects a dynamic mosaic of $C_3$ and $C_4$ plant communities (Table A in S4 File). $C_3$ grasses such as *Festuca*, *Stipa*, and *Agropyrum* dominate in spring and early summer but senesce during the hot, dry season. In contrast, $C_4$ taxa expand in late summer and autumn, including annual grasses (Bothriochloa) and halophytic shrubs. Genera such as *Halostachys* and *Kalidium* are predominantly $C_4$, while *Salsola*, *Suaeda*, and *Halocnemum* include both $C_3$ and $C_4$ species. These woody halophytes often remain green into winter and, together with hardy $C_3$ shrubs like *Capparis spinosa* and *Artemisia fragrans*, would have provided dependable browse when $C_3$ grasses were no longer available. Moreover, local microclimatic conditions, such as saline flats, disturbed soils, and sheltered south-facing slopes, support persistent $C_4$ plant growth even into cooler seasons in the Sharur plain [77,78]. $C_4$ grasslands have also been documented in northwestern Iran until late autumn [79].

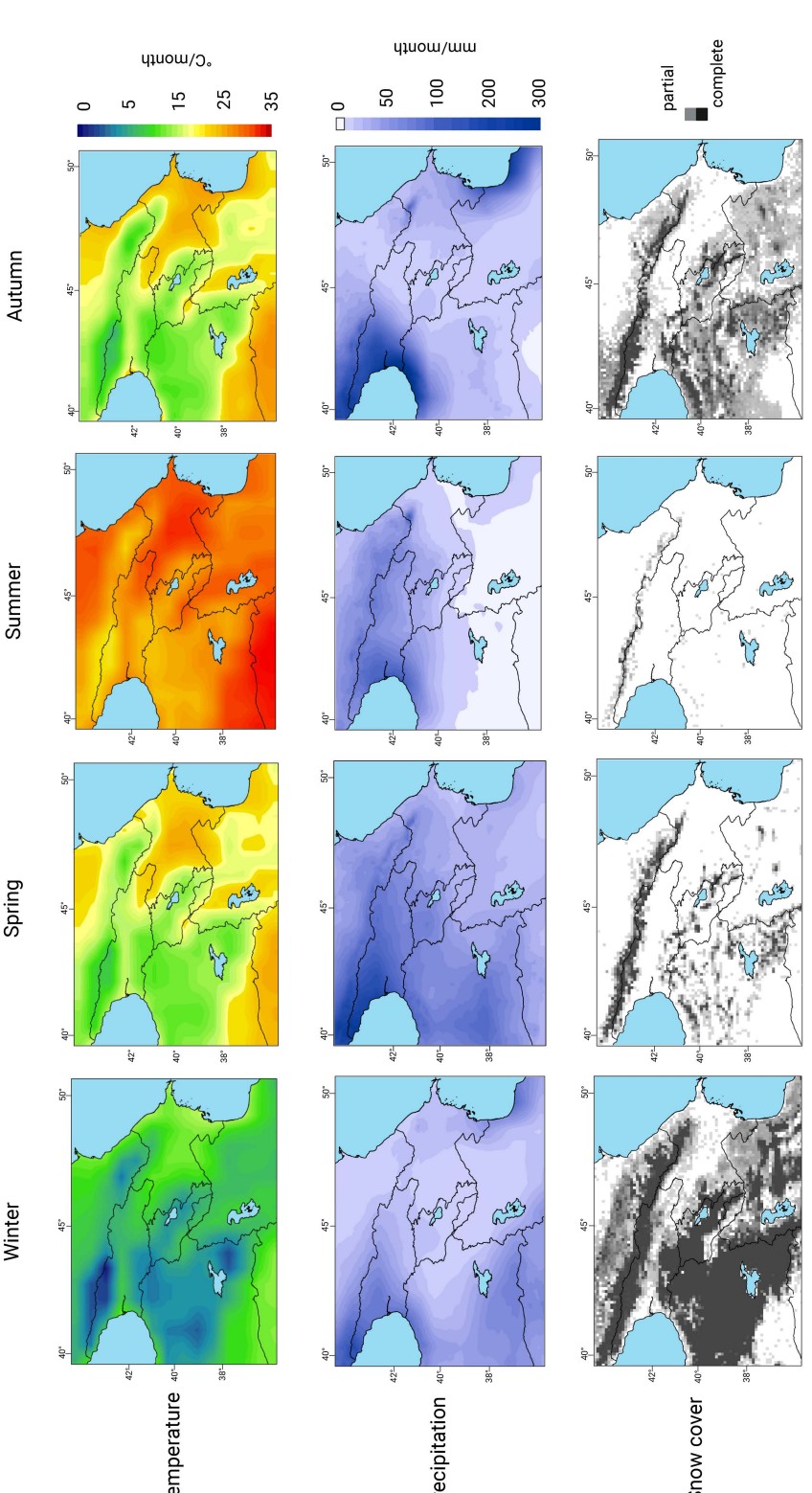

**Fig 2. Spatial variation of temperature, precipitation and snow cover across the Caucasus.** Climate data taken from https://giovanni.gsfc.nasa.gov/giovanni/ and manipulated using ArcMap V10.8.2. Assembled using https://app.biorender.com/.

$\delta^{13}C$ values from modern vegetation in the region also exhibit elevation-linked enrichment [63]. Available palaeoclimatic evidence suggests that Early Bronze Age climate conditions in the region were broadly comparable to those of today [80,81]. Full details on regional climate, vegetation, hydrology, and isotopic variation are provided in S1 File, S3 and S4 Data.

Maxta I is a low mound (1.5–2 m in height) covering 0.78 hectares, although Early Bronze Age materials extend across approximately 3 hectares [82,83]. Excavations confirm that Maxta I was exclusively occupied during the Early Bronze Age (3316–2921 cal. BC) [66].

The animal bones analysed for this study stem from structures 1–4, 5 and 6 (2013 season) (Fig 3a). Eighteen caprine mandibles from Structures 1–4, 5, and 6 (2013) are currently being dated at Oxford. The associated collagen and sequential isotope data are presented here. Several Kura-Araxes sites, including are situated on the Sharur Plain, where settlement density increases along the Arpachay River due to the availability of water and other resources. Maxta I can therefore be considered in relation to these and other excavated Kura–Araxes settlements in Nakhchivan [88–91](Fig 1).

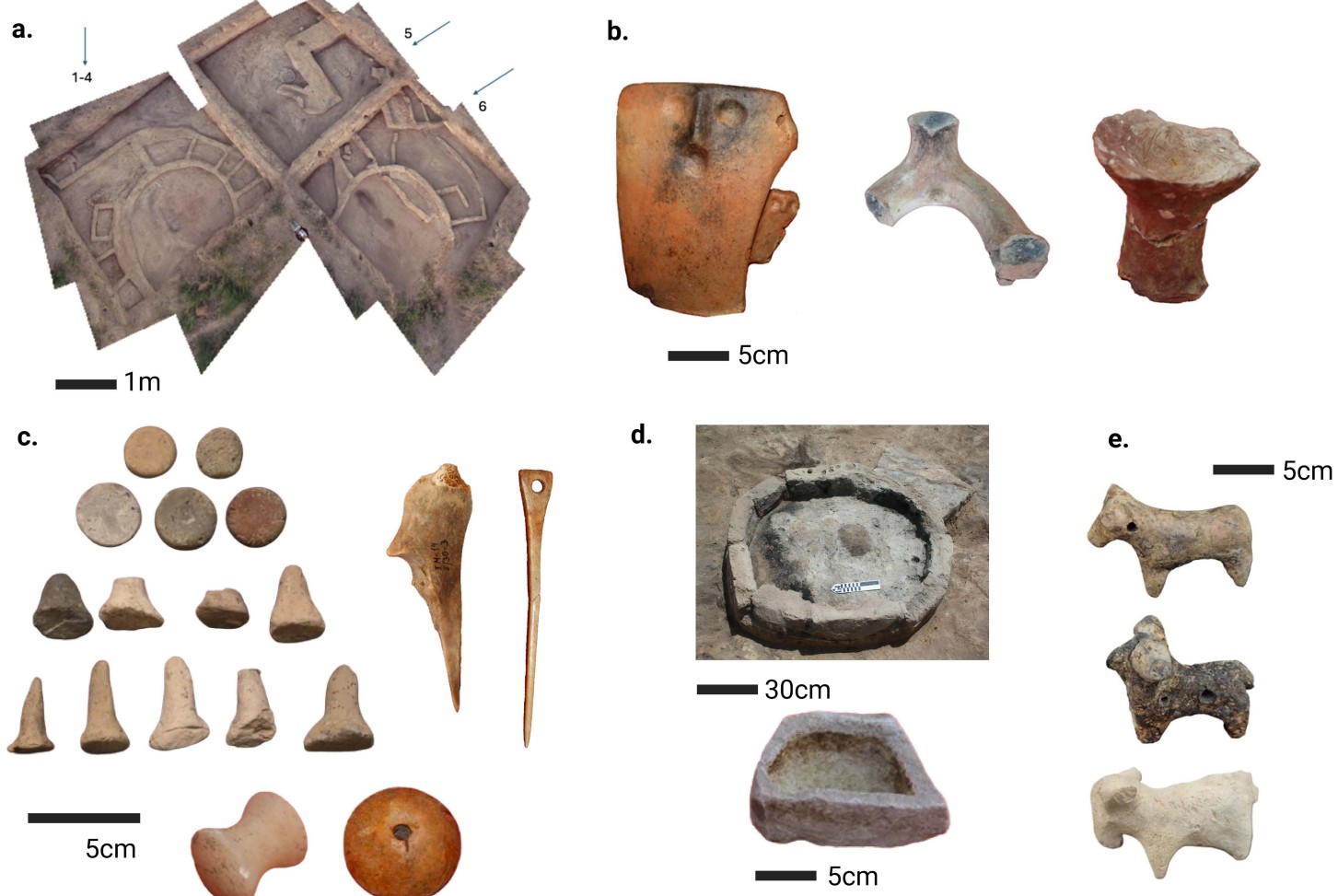

**Fig 3. Settlement data from Maxta I. a.** Architectural plan of structures 1-4, 5 and 6 at Maxta I [84]. **b.** Hearth structures [85, 86]. **c.** Evidence for possible leather and fiber (animal and/or plant) working [84]. **d.** Evidence for metal production [86].**e.** Animal symbolism [87]. Images are courtesy of Safar Ashurov and Fidan Khalafova Aliyeva. Additional images can be found in S2 File.

Maxta I presents a densely organised settlement built primarily with mud-brick and wattle-and-daub. Excavations revealed three main building types: circular houses, circular buildings with rectangular annexes, and freestanding rectangular structures. Stratigraphy shows a transition from circular to more complex architectural plans in later phases, including the addition of ancillary rooms and multi-course walls [68,70]. Comparable sequences are documented at Çaqqallıqtəpə and Serkertepe [92–94]. Structures at Maxta I reflect notable functional variability, ranging from communal to individual household use. Structure 1-4 contained ritual assemblages suggestive of ceremonial activity, while structure 5 featured installations for shared food processing and metalworking. In contrast, structure 6 displayed a more self-contained domestic layout. Hearths and benches consistently appear in central spaces, aligning with broader Kura-Araxes spatial norms [95]. Hearth types include circular floor-set examples, round mangals, and portable horseshoe-shaped andirons [96]. The material culture particularly spans bone tools [84], clay tokens [85], unbaked perforated *asmalars* [84], and metalworking remains [84]. Ceramics are largely undecorated, though some bear volute and animal motifs [84,85]. Thirty-three zoomorphic figurines have been identified [85].

The stable isotope ratios of $\delta^{18}O$, $\delta^{13}C$, and $\delta^{15}N$ in domestic herbivores are shaped by a suite of environmental, ecological, and physiological factors that influence water sources, plant carbon pathways, and nitrogen availability and cycling.

Animal bioapatite $\delta^{18}O$ reflects the isotopic composition of ingested water, both meteoric and plant-derived, and is modified by physiological fractionation [97,98]. Precipitation $\delta^{18}O$ varies with temperature [99–101], seasonality [102], continentality [99], altitude ("rainout"103), and shifting moisture sources. Leaf water $\delta^{18}O$ is further enriched by evapo-transpiration, especially in warm, arid conditions [103–106], so non-obligate drinkers (e.g., sheep, goats) that derive much water from vegetation exhibit higher enamel $\delta^{18}O$ than obligate drinkers (cattle, equids) drinking surface water [107–110]. Browsers and grazers may thus overlap or diverge in $\delta^{18}O$ depending on plant type and rooting depth [111–113], necessitating paired $\delta^{13}C$ analysis to disentangle dietary from hydrological signals.

$\delta^{13}C$ values primarily track the isotopic composition of the animal's diet, especially the proportion of $C_3$ versus $C_4$ plants consumed [114], which is influenced by vegetation type and climatic conditions. In $C_3$ plants, water stress and reduced humidity constrain stomatal conductance and diminish discrimination against $^{13}C$, raising $\delta^{13}C$ values [115–117], whereas $C_4$ species are largely insensitive to soil moisture variations [118,119]. Saline substrates also elevate $\delta^{13}C$ by limiting discrimination [120]. With altitude, lower ambient $CO_2$ partial pressures (Pi/Pa) reduce carboxylation discrimination, yielding a + 0.12–0.46‰ $\delta^{13}C$ enrichment per 100 m [63,121]. Temperature deviations from optimal growth impart only minor $\delta^{13}C$ shifts [122], while seasonal changes in $C_3/C_4$ plant dominance further modulate $\delta^{13}C$ patterns. Under dense canopies, understory plants exhibit more negative $\delta^{13}C$ due to altered light and humidity regimes [123]. Non-vascular cryptogams and fungi display systematically higher $\delta^{13}C$ than vascular plants [124,125], and within vascular taxa, leaves are typically depleted in $^{13}C$ relative to stems and roots [126].

In herbivorous mammals, bone collagen $\delta^{13}C$ and $\delta^{15}N$ values integrate the isotopic signature of consumed plant protein over the lifespan, often spanning decades, because bone is continuously remodelled [127,128]. As a result, collagen $\delta^{13}C$ and $\delta^{15}N$ specifically trace the protein fraction of herbivore diets, reflecting long-term foraging ecology rather than short-term dietary bouts. For non–nitrogen-fixing plants, tissue $\delta^{15}N$ closely tracks the $\delta^{15}N$ of available soil nitrogen, since discrimination against $^{15}N$ is minimal when plant nitrogen demand outpaces supply [129]. Soil $\delta^{15}N$ itself varies with factors such as parent material, pH, disturbance, atmospheric $N_2$ inputs, and land use [130–133]. In warm, dry climates, enhanced $NH_3$ volatilization drives soils to higher $\delta^{15}N$ values [134–136], while rooting depth, uptake of $NO_3^-$ versus $NH_4^+$, turnover rates, and mycorrhizal partnerships further modulate plant $\delta^{15}N$ [137]. Agricultural amendments, notably manure application, can elevate crop $\delta^{15}N$ substantially [132,133,138] and penned livestock concentrate dung deposits, creating persistent high-$\delta^{15}N$ soil patches [139]. In animal tissues, bone collagen $\delta^{15}N$ is enriched by approximately 3–4 ‰ per trophic level [140,141], a pattern also evident in nursing juveniles exhibiting higher values than adults [142,143]. Among herbivores, higher $\delta^{15}N$ in arid habitats reflects the isotopic signature of drought-adapted vegetation rather than physiological stress [144–147]. Nonetheless, starvation or poor-quality diets can raise tissue $\delta^{15}N$ [148,149], and interspecific differences in cycling and digestive physiology may introduce additional variation [139].

## Materials and methods

### Materials

All animal bones were retrieved in the field by hand-picking, or through the picking of post-flotation heavy fraction sediments. The presented postcranial assemblage does not represent the entire animal bone assemblage from Maxta I, but rather a sub-set. The analysed postcranial assemblage includes approximately 25% of the faunal material from each context, focusing on diagnostic elements. Due to time constraints caused by the COVID-19 outbreak, the research trip to Nakhchivan had to be cut short, which limited the opportunity to fully document the entire assemblage. However, all mandibles and tooth wear across all taxa at Maxta I have been thoroughly analysed. Permission to conduct the research was granted by the Department of Archaeology, Azerbaijan National Academy of Sciences, Nakhchivan Branch.

### Zooarchaeology

The identification of the zooarchaeological material was carried out in the field in Nakhchivan in 2020 using osteological manuals and digital resources. The distinction between *Bos taurus* and *Bos primigenius* was made based on size. For the differentiation between postcranial elements of domestic cattle (Bos taurus) and red deer (*Cervus elaphus*) Prummel (1988) [150] criteria were used. Caprines found in the Caucasus in the Early Bronze Age are domestic sheep (*Ovis aries*), domestic goat (*Capra hircus*), wild sheep (*Ovis orientalis*), wild bezoar goat (*Capra aegagrus*), the Caucasian tur; (*Capra caucasica*), the western tur and *Capra cylindricornis*, the eastern tur. Further the Armenian or Asiatic mouflon (*Ovis gmelini gmelini*) is also present in the region. The chamois (*Rupicapra rupicapra*) is also native to northeastern Turkey and the Caucasus. The identification between sheep (*Ovis aries*) and goat (*Capra hircus*) was based on morphological criteria by Boessneck (1969) [151]; Prummel and Frisch (1986) [152] Zeder and Lapham (2010) [153].

The distinction between sheep and goat mandibular teeth was based on Halstead et al. (2002) [154]. To increase accuracy, sheep and goat were only differentiated if at least two clear morphological criteria were present. Caprinae bones which could not be distinguished were assigned to sheep/goat (*Ovis*/*Capra*). Pig remains were assigned to pig/boar, genus: *Sus*, due to insufficient measurements.

In the Caucasus, no domestic equid species have been reported from the Early Bronze Age. Equid species from Early Bronze Age contexts in the Caucasus previously have been identified as wild ass (*Equus hemiones*) or wild horse (*Equus ferus*) [155]. The difficulty in assigning postcranial and dental remains to Equidae species level has been widely discussed [156–158]. Baxter (1998) [159] and Davis (1980) [156] criteria were used to differentiate between horse, onager and wild ass teeth. However, the differentiation between equid species remains contested [160,161].

The cervidae species present in the Caucasus in the Early Bronze Age are red deer (*Cervus elaphus*), Persian fallow deer (*Dama mesopotamica*), European fallow deer *(Dama dama)* and roe deer (*Capreolus capreolus*). The wild bovidae species present are gazelle (*Gazella subgutturosa*), aurochs (*Bos primigenius*) and possibly bison (*Bison bonasus*). Further, the Saiga antelope (*Saiga tatarica*), now critically endangered, was previously found across the Caucasus. Identification was carried out using osteological manuals: Stefano (1995) [162], Lister (1996) [163] and Prummel (1988) [150] for deer; Peters (1989) [164], Helmer and Rocheteau (1994) [165] for gazelle; and Jing et al. (2021) [166] for saiga antelope.

For the morphological identification of carnivore remains the following osteological manuals were consulted: Varela and Rodriguez (2004) [167], Johnson (2015) [168] for *Canis*, *Vulpes* and *Felis* and Walker (1985) [169] for large wild carnivores. Identifications were primarily morphological at this stage, due to time constraints on undertaking metrical analysis, which will be used to confirm taxonomic identifications in future.

The assemblage was analysed for anthropogenic and non-anthropogenic taphonomic alterations. Burning was recorded where identified [170]. Gnawing was recorded where identified [170,171]. Weathering was recorded according to Behrensmeyer (1978) [172], with 0 signifying no weathering, 1 = light weathering and 5 = heavy weathering. Butchery was recorded in absence/presence. Bone completeness was calculated to investigate the effect of post-depositional processes

on the assemblage. Bone completeness was estimated following Marean (1991) [173]. For this astragali of sheep, goats and cattle were used. The effects of density mediated attrition was investigated using Symmons (2005, Table 5) [174] for sheep and goats.

Tooth wear on mandibular teeth for caprines was recorded according to Grant (1982) [175]. Mortality profiles based on tooth wear were created following established age classes by Payne (1973) [176] for sheep and goats. Left and right mandibles were not pair matched unless they showed similar tooth wear and derived from the same context.

### Interpreting pastoral systems using the zooarchaeological record

The framework used to interpret pastoral systems and mobility using the zooarchaeological record are outlined in Table 1 and 2.

### Models for the interpretation of caprine age-at-death data

Blaise (2006)185, Helmer (1992) [187], Helmer et al. (2007) [193] and Vigne and Helmer (2007) [192] produced the following models for sheep and goats based on ethnographic work.

**Type A meat**: Most lambs are culled between 6 months to 1 year of age.

**Type B meat:** Most lambs are culled as sub-adults, at 1–2 years.

**Type A milk:** Unweaned lambs are culled 0–3 months of age.

**Type B milk:** Lambs are weaned and kept apart from the ewes. Females peak culling at 2–4 years when milk productivity decreases.

In wild sheep, the natural weaning age has been reported as 12 months of age [194,195]. However, humans are known to wean lambs earlier than that and reported weaning timing in domestic caprines are reported as short as at 4 months (sheep breeds in the modern Caucasus; see Ulimbashev and Ulimbasheva (2020) [196] or as early as 2 months of age in modern Awassi sheep in the Levant [197]. It has also been shown that winter lambing is associated with longer lactation times than spring lambing. Therefore seasonality and season of birth might influence weaning times [198].

**Fleece model:** High mortality rates of animals beyond the age of 4–6 years. There is, inevitably, some degree of equifinality between varying production strategies that may produce similar survivorship and mortality profiles (e.g., between milking and working in cattle and milking and fleece production in caprines [192].

Table 1. Forms of pastoralism and their main characteristics. Based on Blench 2001 [177].

| Form of Pastoralism | Species Composition | Mobility | Degree of Specialisation | Rationale |
|---|---|---|---|---|
| **Nomadic** | One or a combination of species | Mobile; usually no cultivation (though limited cultivation may occur) | One or various products | Opportunistic system that follows pasture resources |
| **Transhumance** | One or a combination of species | Seasonal mobility – either vertical (lowland–highland) or horizontal (low–high rainfall zones) | One or various products | Exploits seasonal availability of resources; avoids clashes with other agro-pastoral activities |
| **Agro-pastoralism** | Combination of species | Settled or seasonal; animals are kept at or near the site; sometimes managed away by pastoralists or other household members | One or various products | Integrates crop and animal production within a mixed system |
| **Silvo-pastoralism** | One or a combination of species (especially pigs, goats, and donkeys) | Variable; animals are seasonally brought to forests or orchards to graze | One or various products | Managed grazing in woodlands or orchards to utilise understorey vegetation |

**Table 2. Zooarchaeological criteria and expectations for identifying pastoral systems.** Summary of key zooarchaeological criteria used to distinguish pastoral systems (settled, nomadic, and seasonal/transhumant) based on herd composition, mortality and sex profiles, provisioning, and butchery practices. Expectations are derived from comparative ethnographic and zooarchaeological studies [170,176–192].

| Question | Approach | Zooarchaeological expectations | | | |
|---|---|---|---|---|---|
| | | Settled | Nomadic | Seasonal | References |
| Species composition of herd | Taxonomic abundance – NISP | Mixed | Single or mixed (not with pig) | Single or mixed | Blench 2001, Göbel 1997 |
| Target of animal production | Mortality profiles, sex composition, pathology | Focused or broad | Focused but can be broad | Focused or broad | Blaise 2006, Helmer 1992 Helmer & Vigne 2007, Payne 1973 |
| Degree of specialisation | Mortality profiles, sex composition | High or low | High but can be low | High or low | Bates 1973, Beck 1986; Behnke 1980, Black-Michaud 1986; Crabtree 1990, 1996b; Stein 1987, 1989, Halstead 1996, Stein 1987 |
| Mobility & seasonality | Mortality profiles, sex composition | All age classes present | Narrow range of age classes present | Gaps in the age classes present | Halstead 1996, Legge et al. 1991, Mashkour & Abdi 2002 |
| Provisioning | Body part representation | – | – | – | Binford 1978, Gifford-Gonzalez 2018 (pp. 413–430 for an overview), Lyman 1987 |
| Culinary practices | Body part representation & butchery patterns | – | – | – | Binford 1978 |
| **Systems** | | **Agro-pastoral Silvo-pastoral** | **Nomadic** | **Transhumant Agro-pastoral Silvo-pastoral** | |

## ZooMS

A total of 20 sheep/goat mandibles were selected. From the selected specimens, 20 mg of bone or dentine chips were weighed into a 1.5 ml microtube and labelled with the sample number. To clean the samples, each sample was soaked in 100l of 50mM ammonium bi- carbonate (Ambic) and left at ambient temperature overnight. Samples were then centrifuged at 12'000 rpm for 1 min and the supernatant was discarded.

100 ul of 50mM AmBic were added to each sample which was incubated for 1h at 65°C. Next, samples were centrifuged for 1 min at 12'000 rpm. Two tubes were labelled, 'EXT' (extract) and 'SE' (second extract), with 50l of the supernatant transferred to the 'EXT' labelled tube and the remaining 5l to the 'SE' labelled tube. The SE tubes were frozen and retained in case of issues with sample processing. The samples in the 'EXT' labelled microtubes were used in the following step for trypsin digestion.

1ul of trypsin solution (0.4l/ g) was added to the 'EXT' labelled tube. Samples were incubated for 18h at 37°C and then centrifuged for 1 min at 12'000 rpm. 10ul of 0.5% TFA was then added to each sample to stop trypsin digestion. Samples were purified using a C18ZipTip and 100ul of conditioning solution (0.1% TFA in 50:50 acetonitrile:ultrapure water) and 100ul of wash solution (0.1% TFA in ultrapure water). All samples from Maxta I failed to produce a signal due to poor collagen preservation. For this reason, the protocol was repeated using the acid demineralisation step. The destructive acid insoluble protocol was carried out following the method established by Welker et al. (2015) [199]. 500ul of cold (4°C) 0.6M HCl was added to each sample (original bone and dentine clippings) and placed in a fridge for 4 days at 4°C. Samples were then rinsed three times with 200ul of 50 Mm AmBic and the protocol was then continued from the gelatinisation step.

Target plates previously cleaned with methanol were used. CHCA (-cyano-4- hydroxycinnamic acid) diluted in a conditioning solution (5 mg/ml) was used as a matrix solution. The matrix was vortexed for 30 seconds and centrifuged for an additional 30 seconds at 10'000 rpm. Samples were spotted directly on the target plates using 0.4ul of the sample and 0.5ul of the matrix solution. For the calibration standards, 0.3ul of Bruker Peptide Callibration Standard II was spotted with 0.4ul matrix. Plates were left to dry and then stored at the temperature of the MALDI ToF MS room.

MALDI-ToF-MS analyses were carried out at the UCL Institute of Archaeology on a Shimadzu MALDI 8020 instrument in linear mode over a m/z range of 900–4000. A pulsed extract at 3200 was used to improve resolution on the higher mass

range, and up to 2400 laser acquisitions were used per spot. The mass spectra were processed and calibrated using Shimadzu software. Processing included baseline subtraction and peak delimitation. The mass spectra were calibrated against an adjacent standard spot containing eight calibrant peptides (Bruker Peptide Calibration Standard II) of 757–3147 m/z range (Bradykinin 1–7, Angiotensin II, Angiotensin I, Substance P, Bombesin, ACTH clip 1–17, ACTH clip 18–39 and Somatostatin 28) – of which seven were used (1046–3147 m/z range).

The obtained collagen fingerprints were manually inspected for the presence of relevant peptide markers α2 757(+16)-3033.4 or α2 757(+16)-3093.4 [200]. Using mMass v. 5.5.0 [201,202], spectra were processed using baseline correction (precision 15, relative offset 25), smoothing (Savitzky-Golay method with a window size of 0.3 m/z and 2 cycles), and then peakpicked (signal to noise threshold of 3.5, picking height of 75). Sheep and goat exhibit almost identical peptide markers with exception of markers COL1α2 757 (+16; m/z 3017.4, 3033.4 for sheep and m/z 3077.4, 3093.4 for goat [200]) and a recently identified COL1α2 375 (m/z 1154, 2028 and 2044 [203]) that facilitate the identification of caprines.

## Sequential stable isotope analysis

All teeth were dry brushed and cleaned of any remaining dirt with dental tools. They were then rinsed under running water and gently brushed with a toothbrush and left to dry overnight. While still in the jaw, molars were extracted from the bone using a circular hand drill saw. For all mandibular M2 sampled, the cementum on and around the posterior pillar was cleaned off using a Tungsten drill bit. For each sample a 1.5 ml Eppendorf tube was labelled and weighed. Intra-tooth sampling for enamel was done using diamond drill bits (1 mm). Drill bits and any dental tools or scalpel used were cleaned after each sample. It was attempted to sample between 4–6 mg of enamel powder per increment. Each increment sampled the full enamel depth. Samples were pre-treated following Tornero et al. (2013) [204]. They were reacted for 4 hours at room temperature with 0.1M Acetic acid (0.1 ml/mg). Samples were rinsed four times with deionised water and freeze-dried for 1h 30 min.

Tooth enamel samples were analysed for their oxygen and carbon isotope signatures at the Bloomsbury Environmental Isotope Facility (BEIF), University College London as well as at the Stable Isotope Facility at Cardiff University. All values are reported in permil in the Vienna Pee Dee Belemnite notation (VPDB) relative to NBS-19. Results at both laboratories are calibrated against an in-house Carrara marble and the international standards NBS-18, NBS-19. At BEIF, samples were analysed on a Thermo Delta V Plus mass spectrometer interfaced to a Thermo Gas Bench II device via a Conflo IV. Standard and unknown sample material (corresponding to around 120 µg of carbonate) was loaded into Exetainer® borosilicate vials (12 ml) closed with a screw cap with pierceable rubber septum. After being flushed with He, each vial was manually acidified with 100% Phosphoric acid (0.1 ml) using a syringe for injection via the septa. Samples were then left to react for over 1 hour at 70°C to ensure complete equilibration. Precision (1 s.d.) of all standards on signals greater than 1000mv is 0.10‰ for $^{13}$C and 0.12‰ for $^{18}$O. At Cardiff University, the enamel powders were measured on a Thermo Mat 253 dual inlet mass spectrometer coupled to a Kiel IV carbonate preparation device. Samples were acidified for 5 minutes with >100% ortho-phosphoric acid at 70°C. The long term precision of the in-house Carrara marble standard is 0.032‰ in δ$^{18}$O and 0.021‰ in δ$^{13}$C.

Sequential stable isotope analysis targets the enamel mineralisation period, as enamel does not remodel after full mineralisation and thus preserves a chronological isotopic record [205,206]. Enamel mineralisation is a progressive, discontinuous process occurring in successive fronts during the maturation stage, meaning layers are not mineralised simultaneously [207]. In caprines, enamel maturation takes approximately twice as long as matrix secretion [207], introducing a time lag between enamel formation and isotopic signal incorporation. In sheep, full enamel mineralisation spans approximately six months [206]. The carbon (δ$^{13}$C) and oxygen (δ$^{18}$O) isotope ratios of tooth enamel reflect dietary and environmental inputs during the period of enamel mineralisation.

The sample set comprises 10 sheep and goat mandibular M2 molars, yielding a total of 92 measurements. Species identification was conducted using ZooMS. In sheep, the M2 crown begins forming between the first and second month of

life and completes by 12 months [208–210]. In goats, crown formation occurs between the first and second month and is typically complete by 10–13 months [211,212]. There is a delay between enamel formation and isotopic signal incorporation [206,207]. Thus, each M2 provides an isotopic record of ~6–18 months, approximately the first year of life.

In sequential sampling of herbivore tooth enamel, perpendicular lines are drilled from the occlusal surface towards the enamel-root junction (ERJ), which serves as a fixed chronological reference point. The accuracy of reconstructing isotopic variation through time depends on the enamel mineralisation pattern, the direction of mineralisation fronts, and the overall duration of the process [206]. Since enamel mineralises progressively and discontinuously, each sample contains material formed over a range of time, blending earlier and later signals [213]. Nevertheless, a chronological trend remains preserved, allowing for the reconstruction of seasonal isotopic patterns, such as summer peaks and winter troughs [205,206,214].

## Collagen stable isotope analysis

All bone samples were prepared following the modified collagen extraction method of Brock et al. (2010) [215]. The $\delta^{13}C$ and $\delta^{15}N$ ratios of the bone collagen were determined using a Sercon 20–22 continuous-flow isotope ratio mass spectrometer (CF-IRMS) coupled via an elemental analyser (EA) at the Oxford Radiocarbon Accelerator Unit. For every 8 unknown samples, an in-house reference of alanine is analysed, and an additional sample of alanine is analysed to assess reproducibility. Data is scaled against USGS-40 and USGS-41. All samples were analysed twice; once whilst CO2 was collected for radiocarbon dating, and once in a batch dedicated to stable isotope analysis. Data is in good agreement between aliquots. Stable isotope ratios of bone collagen are reported as delta ($\delta$) values relative to an international standard: for carbon isotope ratios in collagen this is Vienna Pee Dee Belemnite (VPDB), and for nitrogen isotope ratios in collagen this is AIR, with precisions of $\pm 0.2$‰ and $\pm 0.3$‰ for $\delta^{13}C$ and $\delta^{15}N$, respectively. Values are reported using the (‰) notation, where $\delta$ (‰) = (Rsample/Rstandard) − 1, and R is the $^{13}C/^{12}C$ or $^{15}N/^{14}N$ ratio.

Bone is continually remodeled throughout life, therefore collagen $\delta^{13}C$ and $\delta^{15}N$ values reflect the dietary protein intake averaged over decades [127], with turnover rates varying by skeletal element [216]. Collagen isotope ratios specifically record the protein fraction of an animal's diet. The age-at-death of specimens selected is outlined in Table A in S6 Data.

## Threshold $\delta^{13}C$ values in tooth enamel and bone collagen

In order to calculate the $\delta^{13}C$ values of $C_3$ and $C_4$ plants in enamel for the Caucasus, we collected published $\delta^{13}C$ values of $C_3$ [63,217] and $C_4$ plants [77] in the Caucasus. These were used to establish regional cut-offs for pure $C_3$ (−11.58 ± 3.00) and pure $C_4$ diets (4.18 ± 1.17) in enamel [218] (Table B in S4 Data). Modern plant $\delta^{13}C$ values were corrected by + 1.5‰ to compensate for the fossil fuel effect [219]. To estimate $\delta^{13}C$ values in enamel, an enamel-diet $\delta^{13}C$-enrichment factor (e*) of +14.1‰ for ruminants (sheep/goats in our study) was applied [218]. The equations according to Cerling and Harris (1999) [218] were used to estimate the $\delta^{13}C$enamel.

We also note that according to Janzen et al. (2023) [57], synthesis of modern and archaeological plant $\delta^{13}C$ data from the South Caucasus suggests that values around −6.7 ‰ in enamel represent the extreme enrichment possible for diets derived entirely from water-stressed $C_3$ plants.

The $\delta^{13}C$ values found in bone collagen typically exceed those in plants by 5%. This can fluctuate because the carbon in collagen primarily originates from the protein in an animal's diet [220,221]. With previously established $C_3$ plant $\delta^{13}C$ values, we expect ~ −20.3‰ for a 100% $C_3$ diet based on regional $\delta^{13}C$ values of $C_3$ plants [63,217] and ~ −6.3‰ for 100% $C_4$ diet, based on $\delta^{13}C$ values of regional $C_4$ plants [77], in herbivore bone (Table B in S4 Data). A limitation of this approach is the absence of modern isotopic measurements of $C_3$ and $C_4$ plants from the immediate vicinity of Maxta, which prevents the establishment of a truly local baseline. As a result, the isotopic thresholds used here rely on the closest available regional proxies and should be interpreted as approximate comparative values rather than fixed local end-members.

## Modelling of δ¹⁸O sequences

To assess birth seasonality, we modeled intra-tooth δ¹⁸O sequences to capture annual cyclical variation in ingested meteoric water, which reflects seasonal climate during tooth formation. Since the timing of tooth development is highly predictable and species-specific [208–212], the position of δ¹⁸O values along the enamel crown provides a chronological proxy for early-life events such as birth. Sequences were fitted using a modified cosine function [222]:

$$\delta^{18}O_m = Ae\{((X - X\_B)/X\_A)\}\cos((2\pi x - x_0)/X + b_x) + M + px$$

where δ¹⁸Om is the modelled δ¹⁸O; x is the distance from the enamel-root junction; X is the period (in mm), or the length of tooth crown potentially formed over a whole annual cycle; A is the amplitude (=max–min/2) (in ‰); x0 is the delay (mm); δ¹⁸O attains maximum value when x = x0; M is the mean (=(max + min)/2) expressed in ‰. Specimens were not modelled that have sequences with a very low amplitude of variation, the absence of a sinusoidal pattern of variation, the absence of a clear maximum or a missing Enamel-Root-Junction (ERJ). The variability in tooth size is eliminated through normalisation of distances using the period X of the δ¹⁸O cycle. The position of the maximum δ¹⁸O values in the tooth crown is therefore expressed as x0/X. Uniform x0/X values across individuals indicate synchronized births, while staggered values reflect a spread of births throughout the year. The resulting x0/X ratio for each specimen is a reference value for the specimen's season of birth. Season of birth is estimated by comparison with reference x0/X ratios obtained in modern sheep and goats [222–228]. Results from the modelling of the δ¹⁸O sequences measured in Maxta I M2s are shown in S7 Data. All results are shown using a circular representation to reflect the cyclical nature of seasonality.

## Phase shift modelling

Phase shift between the oxygen and carbon sequences were estimated out where the δ¹⁸O and δ¹³C sequences are processed using a sinusoidal model approach [229] (S8 Data). Although the anticipated pattern is not strictly sinusoidal, it is assumed to be periodic with the same period for both signals, corresponding to a full year. Data were fitted by the following model:

$$\delta^{18}O(d) = A\delta^{18}O \sin(2n/D\, d + \Psi\delta^{18}O) + M\delta^{18}O$$

$$\delta^{13}C(d) = A\delta^{13}C \sin(2n/D\, d + \Psi\delta^{13}C) + M\delta^{13}C$$

## Results

### Zooarchaeology

**Taphonomic analysis.** In total, 410 (NISP) animal bones were analysed from the EBA layers at the site of Maxta I. Natural and human induced taphonomic agents affect the zooarchaeological assemblage to a moderate degree (Fig 4b). Overall, burning only affects 5% of the total assemblage at Maxta I. Signs of gnawing are very uncommon among the assemblage (0.2%). The majority of the assemblage is affected by weathering, however only to a mild degree (stage 1 and 2). Signs of butchery are common within the assemblage, affecting 17% of the total assemblage. The anatomical profile of caprine remains is less complete and dominated by mandibles and humeri, whereas cattle are represented mainly by head, shoulder, and haunch elements, with nearly all skeletal parts present except for sections of the lower limbs (Table E in S5 Data). The effect of density mediated attrition among cattle and caprine bones at Maxta I was investigated. In caprines and cattle, density mediated attrition has no effect on the preservation of skeletal elements within the assemblage (Fig 4c). The completeness [173] for tarsal bones (astragali) at Maxta I is 100%. This suggests the assemblage did not suffer from any significant post-depositional decay. The taphonomic analysis is further summarised in Table A, B and C in S5 Data.

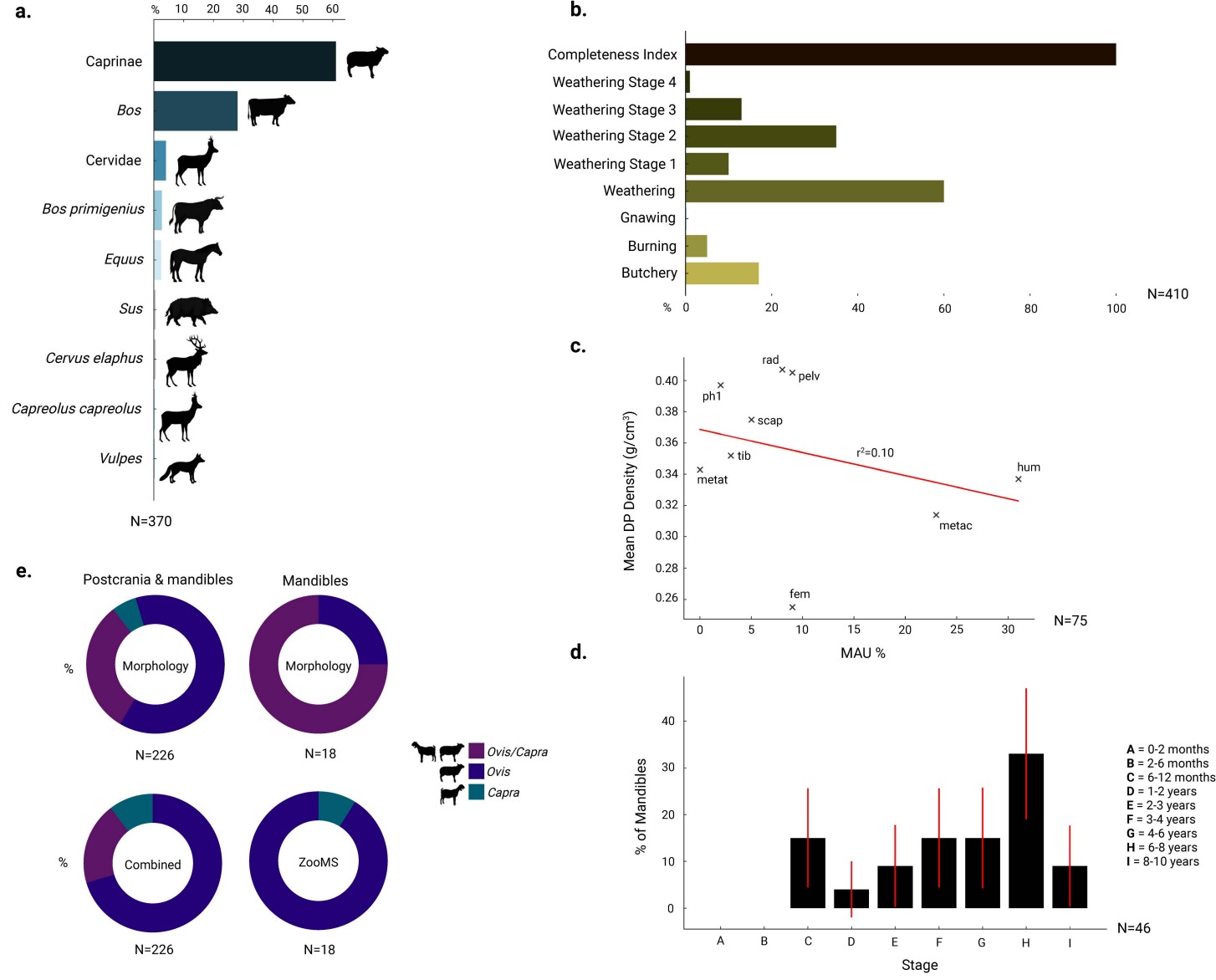

**Fig 4. Zooarchaeological and ZooMS analysis of Maxta I. a.** Taxonomic abundance (NISP %). **b.** Multi-variate taphonomic analysis of the zooarchaeological assemblage. Detailed information about the taphonomic analysis is presented in Table A, B & C in S5 Data. **c.** Correlation between MAU % of caprine (*Ovis/Capra*) bones and Symmons' bone density [174]. **d.** Caprine (*Ovis/Capra*) mortality profile based on dental wear [176], associated with credibility intervals (Dirichlet distribution law [230]). **e.** Identification of *Ovis* and *Capra* mandibles and postcranial elements using morphological and ZooMS analyses. ZooMS improved genus-level resolution of previously classified *Ovis/Capra* remains. The top panels show morphological identifications for all specimens (left, N = 226) and for the subset of mandibles analysed by ZooMS (right, N = 20). The bottom panels show ZooMS identifications for these 20 mandibles (right) and combined results for the entire assemblage (left, N = 226), where molecular identifications were integrated with morphological observations to refine overall taxonomic assignments. Colours indicate *Ovis/Capra* (purple), *Ovis* (blue), and *Capra* (teal).

**Taxonomic abundance.** The zooarchaeological analysis is summarised in Table D in S5 Data and Fig 4a. Maxta I is dominated by caprines, followed by cattle. Suids are very rare at Maxta I, only accounting for less than 1% of the domestic assemblage (if these indeed represent domestic pigs). Sheep dominate over goats at the site. Interesting is also the relatively high contribution of wild ungulates to the assemblage, accounting for 10% of the identifiable assemblage. These consist of wild horse, aurochs and deer species.

**Age-at-death.** Mortality profiles using dental wear in caprines, suggests some young specimens are killed-off between 6–12 months as well to a lesser degree between 1–3 years of age at the site (Fig 4d). Age classes C-I are present, while the youngest age classes A-B are absent. However, the majority of individuals fall into older age ranges between 3–8 years of age. This is also suggested by the epiphyseal fusion data, where the majority of caprines are kept past the penultimate fusion stage E (30–48 + months) (Table G in S5 Data). Cattle mortality profiles, based on small fusion and dental wear samples, indicate that most individuals were culled between 0–28 months (Classes B–D); a few very young animals (<10 months, Stage A) are present but may be biased by sample size, and few survived beyond the final fusion stage (>42–48 months) (Table F in S5 Data).

## ZooMS

All samples (18) from Maxta I, failed to produce a signal using the non-destructive protocol [231]. After applying the destructive acid insoluble method [199], all 18 samples rendered spectra. Out of the 18 sampled mandibles, 17 were identified as *Ovis* and a single specimen as *Capra*, based on the peptide marker: COL 1α2 757(+16). ZooMS enabled the secure genus-level identification of *Ovis/Capra* stable isotope sampled mandibles, which were previously classified only as caprines (Fig 4e). We note that Table A in S6 Data lists only the specimens used for sequential enamel sampling and/or collagen analyses. ZooMS was performed on a wider set of mandibles, and therefore the total number of ZooMS-analysed specimens does not fully overlap with those listed in Table A. The full ZooMS table and spectra are available on Mendeley Data https://doi.org/10.17632/24dyjf7c5j.1.

## Stable isotope analysis

**Sequential enamel bioapatite δ¹⁸O and δ¹³C values.** The results of the sequential stable isotope analysis on caprines (*Ovis/Capra*) are summarised in Table D in S6 Data and Fig 5. The $\delta^{18}O$ values of caprines from Maxta I range from −10.2‰ to 0.6‰, with an intra-tooth amplitude (when optima are identifiable) of 4.7‰ to 8.9‰ (Table E in S6 Data). Specimens MI16, MI18, MI21, MI25, MI42, MI57 and MI59 show a sinusoidal variation in their oxygen sequences. However, specimens MI46, MI53 and MI57 do not show a clear minimum (Fig 5). These were thereby excluded from the statistical analysis. The $\delta^{18}O$ mid-point values are not different at Maxta I (single factor ANOVA, p-value: 1.0). The variation in intra-tooth amplitude is similar between specimens at Maxta I, however MI16 and MI21 are displaying a comparatively reduced amplitude. All Maxta I specimens fall within the expected range of predicted local precipitation [75] (Table A in S3 Data). We thereby interpret the maximum $\delta^{18}O$ values as the summer signal and the minimum $\delta^{18}O$ as the winter signal. The $\delta^{13}C$ values of caprines from Maxta I range from −10.3‰ to −4.2‰, with an intra-tooth amplitude of 1.2‰ to 4.9‰ (when optima are identifiable). Specimen MI47 shows a clear sinusoidal variation in the $\delta^{13}C$ curve. Specimens MI42, MI59, MI18, MI21 display dampened sinusoidal variation. Specimens MI16 and MI25 do not show clear sinusoidal variation in $\delta^{13}C$ but a minimum and maximum are identifiable using the oxygen sequence. MI46, MI53 and MI57 do not show a clear maximum (Fig 5). These were therefore excluded from any statistical analysis. The mid-point $\delta^{13}C$ values of caprines at Maxta I are not significantly different (single factor ANOVA, p-value: 0.7). Specimens MI42, MI18, MI21, MI59 and MI16 show a reduced amplitude, whereas MI53 and MI47 show larger amplitudes. One pattern (A) is identifiable among the sequential isotope profiles from Maxta I with three variations (A.1, A.2 and A.3).

Variation A.1 is represented by specimens MI42, MI18, and MI59, all of which exhibit sinusoidal variation in their oxygen isotope sequences. Sinusoidal variation, although significantly dampened, is also visible in the carbon sequence. The oxygen and carbon sequences vary in opposition (phase shift: 197°: see methods). Summer $\delta^{18}O$ values are somewhat depleted compared to expected values for lowland sites in the region (Table B in S3 Data). However, the depletion is not substantial enough to indicate a dominant contribution from highly depleted sources, such as snowmelt. Altitude appears to have little to no effect on $\delta^{18}O$ values in the highlands of the Caucasus [76]. Thus, any potential influence of altitude or snowmelt on $\delta^{18}O$ values might have been overridden by the temperature effect [76]. During spring and summer, caprines

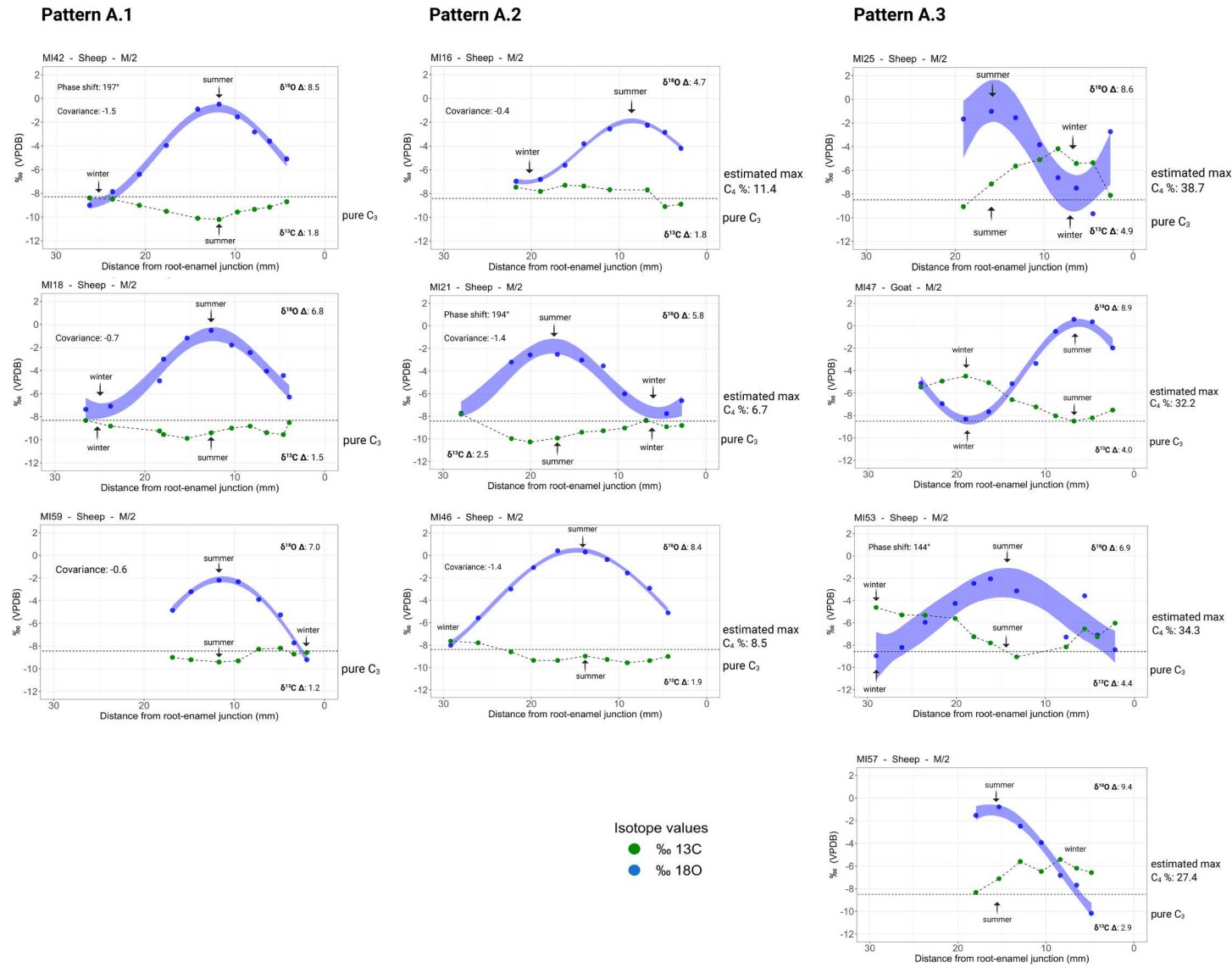

**Fig 5. Results from sequential stable carbon (δ¹³C) and oxygen (δ¹⁸O) isotope analysis of caprines (N = 10) from Maxta I.**

likely consumed a mix of water sources, including enriched leaf water, snowmelt, and occasional late-afternoon summer precipitation which is common in the highlands [232]. Winter (minimum) δ¹⁸O values across all specimens reflect typical seasonal lows expected for the Sharur plain (Table A in S3 Data). The carbon isotope sequences indicate a consistently pure $C_3$ diet throughout the seasonal cycle (Fig 5). Winter $C_3$ values are comparatively enriched, while summer values reflect the consumption of freshly grown $C_3$ vegetation. Given the semi-arid and arid steppe environment surrounding Maxta I, we would expect the presence of $C_4$ grasses throughout the summer. The absence of a $C_4$ signal in the isotope record suggests the seasonal movement of animals to a pure $C_3$ environment from early summer through autumn, such as a surrounding highland environment. Subalpine and alpine zones in the Caucasus are dominated by lush $C_3$ meadows during spring and summer, supporting this interpretation. The comparatively enriched winter $C_3$ values likely reflect the consumption of water-stressed $C_3$ vegetation on the lowland plains near the site. Winters at Maxta I are cold and arid,

conditions known to increase $\delta^{13}C$ values in plants due to both reduced stomatal conductance and photosynthetic stress [116]. Additionally, the saline soils of the Sharur plain may further enrich $\delta^{13}C$ values [120]. It is also plausible that, during snow-covered periods, animals fed on woody twigs, which are naturally enriched in $\delta^{13}C$[129].

Variation A.2 is represented by specimens MI16, MI46, and MI21. Specimens MI16 and MI21 exhibit sinusoidal variation in both their oxygen and carbon isotope sequences, while MI46 lacks a clear minimum in its oxygen sequence and a distinct maximum in its carbon sequence. The oxygen and carbon values vary in opposition, with a phase shift of 194° (see method/ S8 Data). This pattern also reveals a small but consistent $C_4$ component, ranging from 7% to 11% in winter. As with Variation A.1, we interpret the winter signal as indicative of free grazing on $C_3$ plants, shrubs and twigs in the Sharur plain. The presence of a minor $C_4$ signal in winter may be explained by the survival of some $C_4$ grasses in rock crevices throughout the cold season [57,58]. In early summer, the carbon values indicate a return to a pure $C_3$ diet, suggesting seasonal movement to highland pastures. There, the animals likely consumed a mix of water sources, including snowmelt, contributing to the slightly lower-than-expected $\delta^{18}O$ summer maximum values in MI16 and MI21. In contrast, specimen MI46 displays $\delta^{18}O$ maxima consistent with expected summer values, possibly reflecting water intake primarily from summer precipitation and enriched leaf water.

Variation A.3 is represented by specimens MI25, MI47, MI53, and MI57. Specimens MI25 and MI47 exhibit sinusoidal variation in their oxygen isotope sequences, while MI53 and MI57 lack a clear minimum. Similarly, MI25 and MI47 show sinusoidal variation in their carbon isotope sequences, whereas MI53 and MI57 lack a distinct maximum. The oxygen and carbon sequences vary in opposition, with a phase shift ranging from 144° to 168° (See S8 Data). This variation is marked by a significant $C_4$ dietary component, ranging from 27% to 39%. We propose that these caprines graze on a mix of dry $C_3$ vegetation and $C_4$ plants in the Sharur plain during winter. Summer-harvested $C_4$ grasses may have been stored and fed to animals during winter, although there is no direct evidence for this practice. However, traces of hay or straw have been identified in the composition of mudbricks used in house construction [85].

In late spring and summer, the carbon isotope values indicate movement to highland pastures, where the animals graze on pure $C_3$ vegetation typical of mountain biomes. Overall, the anti-phase relationship observed between the oxygen and carbon isotope sequences, where carbon maxima aligns with oxygen minima, and vice versa, is best explained by seasonal access to a pure $C_3$ diet during summer. Such a diet would only be available in high-altitude environments in the area during this season. Year-round management of caprines in the lowlands around Maxta I appears unlikely. If this were the case, and assuming normal seasonal variation in $\delta^{18}O$ values, caprines would have had to entirely avoid consuming $C_4$ plants during summer, despite their known prevalence in the Sharur plain. An alternative scenario, where summer and winter isotope signatures are reversed due to consumption of highly depleted snowmelt, is also implausible. All recorded maximum $\delta^{18}O$ values are too high to reflect typical winter conditions (Table A in S3 Data).

**Seasonality of birth.** The results from modelling [222] $\delta^{18}O$ sequences of caprines from Maxta I are shown in S7 Data and Fig 6. Sheep at Maxta I have a cycle period (X) varying between 18.4 and 28.4 mm, reflecting variability in tooth size and highlighting the need to normalize x0 distances by X. The (x0/X) ratios vary from 0.35 to 0.85 in the M2, indicating births in spring and in autumn. The goat, specimen MI47, indicates a late autumn birth. For MI16, MI18 and MI42, the birth-season estimates should be regarded as tentative because the sampled sequences may not capture a true $\delta^{18}O$ minimum.

**Collagen stable $\delta^{13}C$ and $\delta^{15}N$ values.** Eighteen caprine bone samples from Maxta I were analysed for collagen stable isotopes (Fig 7). Mandibles selected for collagen isotope analysis were those with associated sequential enamel isotope records, allowing us to link seasonal practices directly to collagen $\delta^{13}C$ and $\delta^{15}N$ values. Additional specimens lacking sequential data were analysed but cannot be securely assigned to seasonal strategies.

C:N atomic ratios ranged from 3.19 to 3.27, indicating good collagen preservation [234]. %C values range from 42% to 45%. %N values range from 15% to 16%. Collagen yields range from 1.1 to 11wt% (Table B in S6 Data). $\delta^{13}C$ values varied from −19.1‰ to −16.3‰ (mean −17.7 ± 0.8‰), suggesting mixed $C_3/C_4$ diets in all individuals. $C_3$ plants

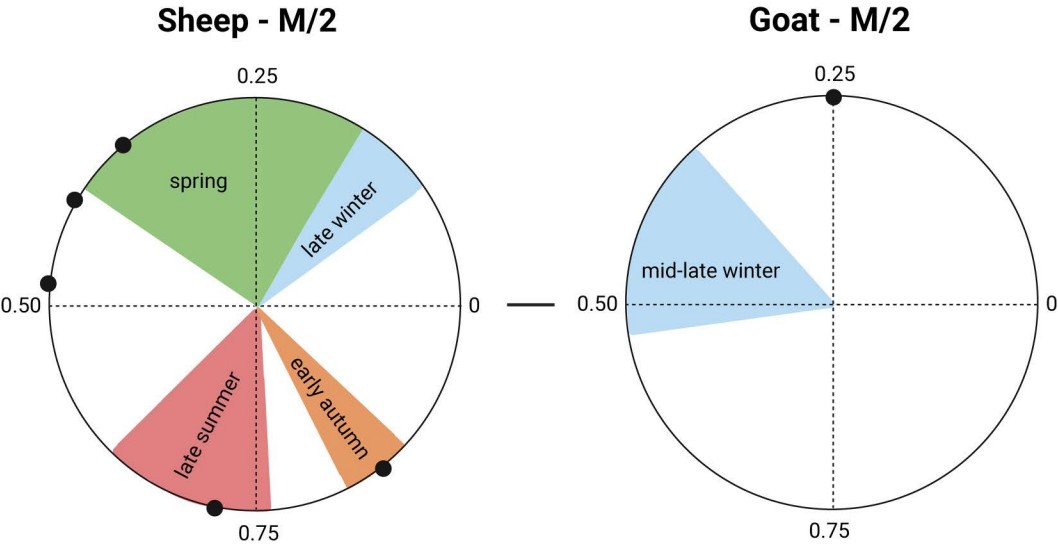

**Fig 6. Distribution of sheep and goat births at Maxta I, as reflected by the position of the maximum δ18O value in the tooth crown (x0), normalized to the period of the cycle (X).** The birth season is compared with modern reference sets. For sheep: Carmejane (CAR) [223], Rousay (ROU) [222], Selgua (XT) [225], Kemenez (KMZ) [226], North Ronaldsay (NR) [227], Le Merle (MRT), and La Fage (MUT) [228]; for goat: Sagalossos [224]. Green, blue, orange, and red shaded areas represent normalised seasonal ranges derived from modern specimens. Archaeological specimens are represented by dots. Detailed information modelled archaeological specimens is provided in S7 Data.

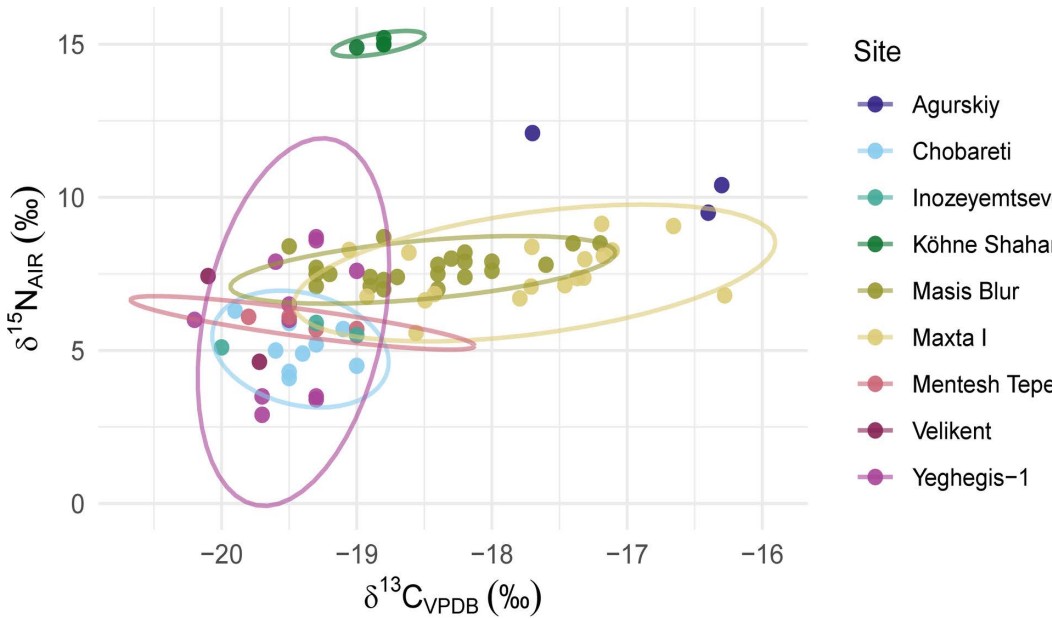

**Fig 7. Bone collagen δ13C and δ15N values of sheep and goats from Maxta I (N = 18; this study), Koehne Shahar [63], Masis Blur [57], Yeghegis-1 [59], Chobareti [53], Mentesh Tepe [50], Agurskiy [233], Inozeyemtsevo [233] and Velikent [233] with 95% confidence ellipses for each site.** Outliers were first removed from the data using the interquartile range (IQR) method, with values outside 1.5 times the IQR from the lower and upper quartiles treated as outliers. Data accessible in S6 Data.

intake dominates (74.6–91.2%), with minor $C_4$ contributions (8.8–25.4%). $\delta^{15}N$ values ranged from 5.6‰ to 9.1‰ (mean 7.5±0.9‰). Statistical analysis revealed no significant correlation between $\delta^{13}C$ and $\delta^{15}N$ values (Pearson r = 0.10, p = 0.68; Spearman ρ = 0.07, p = 0.80), suggesting independent variation in carbon and nitrogen isotopic signatures.

## Discussion

The integration of zooarchaeological, isotopic, archaeobotanical, and settlement data from Maxta I reveals a seasonally adaptive agro-pastoral system that challenges traditional models of Kura-Araxes lifeways. Rather than conforming to a binary classification of nomadic versus sedentary, Maxta I operated through a dual, modular strategy that combined architectural permanence with selective seasonal mobility. Evidence suggests that while part of the community and their herds engaged in summer transhumance to upland pastures, others remained at the site year-round. This interpretation is supported by multiple strands of data: the continuity and rebuilding of domestic structures, the presence of fixed hearths and storage installations, and the demographic profile of livestock indicating year-round caprine presence. Together, these elements reflect a flexible settlement system embedded within a cyclical landscape of movement and resource management. The seasonal coordination of herd mobility, birthing schedules, and secondary product use is illustrated in the proposed model (Fig 8).

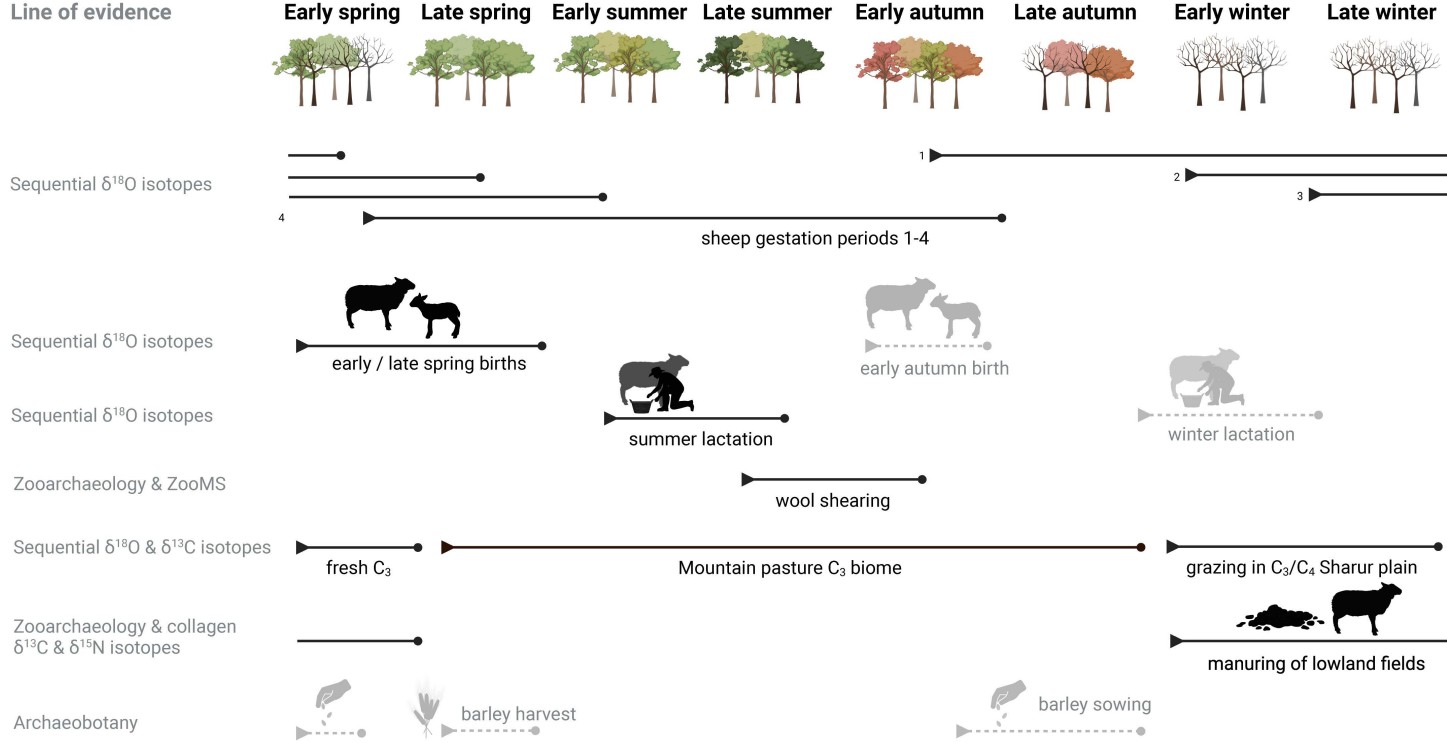

**Fig 8. Schematic representation of possible seasonal pastoral strategies at Maxta I.** This schematic summarises the interplay between herd mobility, dual birthing seasons, architectural permanence, and resource use across the annual cycle. Drawing on stable isotope sequences, zooarchaeological data, and settlement features, the model highlights how caprines were seasonally moved between lowland and highland pastures, with distinct birthing and culling strategies tailored to fleece, meat, and dung production. Permanent domestic infrastructure at Maxta I anchored this flexible, seasonal economy. Autumn births, winter lactation and archaeobotanical interpretations are presented as hypothetical scenarios that align with the available data but cannot be directly demonstrated due to the limited sample size (indicated with dashed lines and lighter colouring). The figure was created using https://app.biorender.com/.

The faunal assemblage at Maxta I is dominated by livestock and in particular sheep, followed by goats and cattle, with an almost complete absence of suids. The significant presence of wild taxa such as onager, red deer, and aurochs suggests that hunting played a strategic role in supplementing the pastoral economy. This pattern may indicate a herding strategy focused on the long-term management of livestock for secondary products (e.g., milk, wool/hair, or traction), in which animals were retained rather than slaughtered for immediate consumption. The absence of neonatal caprines (age classes A and B) implies that birthing occurred away from the site likely in upland pastures while the presence of all post-weaning age classes (C–I), based on the zooarchaeological data, supports the idea that some animals remained locally throughout the year. This points to a dual system of caprine herd management: seasonal movement for some, stationary care for others. Furthermore, cattle seem to have been locally bred (class B), but the overall narrow age range is consistent with a seasonal management model.

Stable isotope analysis confirms this seasonal transhumance, with caprines moving to highland pastures between early summer and late autumn. Yet, architectural evidence firmly anchors the community to the site. There is clear evidence of standardisation and repetition in building practices at Maxta I across successive construction phases. The consistent use of similar building materials and techniques, particularly in the layering of mudbrick structures, indicates the reuse of architectural elements and continuity in spatial organisation [94,86,87]. Some buildings appear to have undergone multi-functional use over time [94,86,87]. Features such as the thickness of accumulated deposits, superimposed hearths, and storage pits located beneath later hearth installations point to long-term, intensive occupation. In several cases, the central hearth was carefully maintained or rebuilt in the same location, reinforcing the persistence of internal spatial layouts over generations [235]. Most KAC settlements appear on virgin or long-abandoned ground to assert a fresh group identity [21,95], Maxta stands out for its repeated reoccupation of the same locality, suggesting a strong sense of place attachment. Maxta I shares key spatial characteristics with other Early Bronze Age settlements in the region, including Yanik Tepe and Shengavit, such as densely packed architecture, wall-sharing between structures, and the absence of formal circulation routes. These features point to a pattern of organically developed, long-lived settlements in Nakhchivan, the Urmia basin, Eastern Anatolia and the Ararat Valley [26,27,88,235].

Industrial installations such as kilns, furnaces, and storage facilities further reflect long-term occupation and domestic investment at Maxta83. The coexistence of permanent architecture with seasonal herd mobility supports a model in which the settlement served as a year-round base for part of the population, functioning simultaneously as a center of domestic, ritual, and productive activity.

Hearth installations at Maxta I reflect this duality: fixed hearths embedded in plastered floors, bordered with mudbrick, and reused across stratigraphic layers suggest long-term spatial continuity, while portable horseshoe-shaped andirons – by some [5,83] associated with mobile lifeways – coexist within the same architectural contexts. However, andirons are more accurately understood not as portable hearths, but as pot supports or functional mediators between cooking and consumption, and can be present in both mobile and sedentary Kura-Araxes settings [236]. Their consistent placement, often accompanied by ritual objects such as figurines, miniature vessels, and animal bones, reinforces the idea of long-term domestic and symbolic continuity [20,237]. These fixed features, central to heating and cooking during cold seasons, strongly suggest that the site was inhabited in winter, and likely maintained during the summer by a subset of the community [237,238]. Additional domestic features including grinding stones [85] and large storage vessels84 indicate on-site food processing and year-round provisioning, reinforcing the integration of seasonal herd management with permanent household routines. This material evidence aligns with a modular system in which some individuals engaged in transhumance, while others such as craft specialists remained at Maxta I.

Seasonal birthing patterns of caprines further support this interpretation. Sheep births at Maxta I are mainly concentrated around spring with isolated births in late summer/early autumn. It is difficult to assess the extent of autumn births due to the small sample size. The gap between these seasonal births might be an intentional strategy employed by

Kura-Araxes herders at Maxta I to focus on not one continuous birthing season but two separate seasons. Spring-born lambs would mature and ewes recover before early summer movement, while autumn births correspond with the return from highland pastures, allowing lambs to acquire sufficient weight before the harsh winters. The modern Armenian mouflon gives birth in early spring which is closely tied to the altitudinal movement of the species [239]. The European mouflon has a recorded birth season of 1–2 months [240]. The extension of birthing seasons through human intervention has been previously archaeologically demonstrated [241–243]. This timing of birth seasons also would have had implications for milk production: spring births enabled lactation in upland pastures throughout summer, while autumn births extended milk availability into the late winter and early spring while herders were in the lowlands (Fig 7). However, the limited dataset prevents assessment of whether the pattern represents deliberate staggering of births or occasional out of season births. If present, such a strategy would have facilitated a more continuous milk supply throughout the year.

However, the kill-off profiles suggest that dairy production was not the primary focus of caprine exploitation at Maxta I. While some milk may have been opportunistically used, the overrepresentation of mature individuals (G–I) in the assemblage, indicates that herd management decisions were driven primarily by fleece exploitation rather than dairying. This interpretation is further supported by lipid residue analysis of Kura-Araxes ceramics, which suggests that while dairying was practiced at some sites, it was not a dominant element of subsistence in the Southern Caucasus in the Early Bronze Age [244]. The focus on wool/hair production is somewhat supported by the material culture from Maxta I discussed earlier (Fig 3). However, slaughter profiles on their own cannot provide sufficient grounds for inferring a focus on secondary products. Comparative evidence from Bronze age contexts in the southern Caucasus likewise points to the exploitation of both plant- and wool or goat hair-based textiles [245–247].

Further, Maxta I yielded numerous awls, needles, pins, borers and spindle whorls which may indicate some degree of involvement in textile-related production activities, though this interpretation remains tentative. [84] (S2 and Fig 3c). Although textile production cannot be excluded as a possible function, ulna bone tools can be employed in a wide range of activities, including hide piercing, fibre pressing, bark splitting, and fish processing [248]. In this light, the Maxta I ulna awl specimens are best understood as multipurpose implements, with leather and fibre working whether plant- or animal-based among several plausible associated uses. Comparable awls are also attested at other KAC sites, in eastern Azerbaijan, Iran [249].

The presence of clay tokens found only at Maxta I and Çaqqallıqtəpə has prompted debate over their function: some interpret them as record-keeping tools [85], while others suggest they served as loom weights [250], could potentially align with textile-related activities. The targeted management strategy at Maxta I is also evident in herd culling patterns: spring-born sheep were often kept to older ages (4–7+years) for possible fleece production, while autumn-born individuals were culled earlier, likely for meat (S7 Data).

The retention of older caprines at Maxta I might suggest their important role in manure production, a vital resource in early agropastoral systems for both fuel and soil enrichment [251]. The presence of all post-weaning age classes (C–I), based on the zooarchaeological record, indicates that a portion of the herd remained at the site year-round, contributing to dung accumulation in and around the settlement. This pattern aligns with seasonal herd mobility, as livestock likely returned to the lowlands in late autumn and winter, coinciding with the need to fertilize fields before sowing. The collagen $\delta^{15}N$ values from Maxta I sheep/goats (6.8‰–9.2‰) exceed the typical range for European herbivores (2.2‰–6.5‰) [252], approaching those of regional carnivores (8.3‰–10.5‰) [52,56]. Compared to other regional sites, Maxta I shows elevated $\delta^{15}N$ relative to highland Chobareti (4.1‰–6.3‰) [52], lowland Mentesh Tepe (5.7‰–7.4‰) [50], and the humid North Caucasus sites of Velikent and Inozyemtsevo [233]. However, the values are consistent with those from lowland Masis Blur (7.0‰–11.8‰) [57], highland Yeghegis-1 (2‰–10‰) [59], and arid environments like Aygurskiy in the North Caucasus [233]. These elevated nitrogen values could reflect prolonged grazing in nitrogen-enriched environments, possibly due to arid climatic conditions, use of manured pastures, or a combination of both. The isotopic evidence suggests intentional or opportunistic use of intensively managed or fertilised grazing areas within the Maxta I herding strategy.

Archaeobotanical evidence [66] from Maxta I is based on a small, spatially restricted sample and should be regarded as preliminary. The assemblage is dominated by barley (*Hordeum vulgare*), with minor bread/macaroni wheat (*Triticum aestivum*/*durum*), small quantities of wild and weedy taxa (e.g., Boraginaceae, *Galium*), and only limited evidence for the pulse bitter vetch (*Vicia ervilia*); no $C_4$ crops have been recovered. Barley was probably autumn-sown [253], a practice that could have benefited from winter dung deposition. Together with the isotopic results, this pattern is consistent with a possible synchronization of pastoral and agricultural cycles: caprines may have been moved to higher pastures after the spring growing season and early summer harvest, when lowland fields had dried to stubble and fresh forage was available in the highlands, and returned in autumn to fertilise fields. However, this scenario remains tentative and requires confirmation through broader archaeobotanical and plant isotopic evidence.

While high mobility has long been assumed for Kura-Araxes communities, direct evidence remains rare. Only Maxta I and Köhne Shahar [63] provide stable isotope data confirming seasonal movement, and in both cases, it reflects local, short-range transhumance rather than long-distance migration. These findings challenge traditional models based on environmental determinism and ethnographic analogy, which often inferred mobility in the absence of direct archaeological evidence [32,36–38].

Mobility at Maxta I should instead be understood in relation to seasonal social organization. In their seminal study of Arctic societies, Mauss and Beuchat [254] described a "dual morphology," in which winter promotes communal, sedentary life and summer encourages dispersed, individual or household-based organization. Similar seasonal reversals and flexibility have been documented elsewhere [255,256]. Studies from Central Asia similarly demonstrate that patterns of mobility and their associated social structures vary significantly in response to ecological and social conditions [188,257]. Such modularity appears to be the rule rather than the exception. Maxta I likely followed a comparable rhythm: winter may have fostered collective domestic life centered around permanent hearths and ritual activity, while summer transhumance involved more mobile, household-based pastoral labour. Palumbi (2016) [258] suggests that Kura-Araxes households exemplified flexibility, interweaving farming and herding so that sites could shift between seasonal camps, permanent villages, or even abandonment over time. Maxta's shifting architecture, alternating between circular and rectangular plans and subdividing houses into smaller units, also reveals a broader cultural and social flexibility. The model of dual morphology questions the static categories often used to define the Kura-Araxes complex – terms such as "mobile," "egalitarian," "uncomplex," or "conservative", many of which derive from outdated social evolutionary or Marxist frameworks [259–261]. These frameworks tend to oversimplify the dynamic realities of early agro-pastoral societies. Recent scholarship has emphasised that social and economic organisation in early societies was not fixed but actively and creatively restructured [262–265].

Frachetti's model [266] of "multiregional non-uniformity" offers a particularly useful lens here: shared cultural features such as red-black ceramics or funerary traditions could coexist with highly diverse and locally embedded lifeways. Maxta I exemplifies this kind of flexibility. Its cultural coherence with the broader Kura-Araxes material tradition masks a local adaptation based on ecological responsiveness, seasonal planning, and social modularity.

Such flexibility was likely not a byproduct of the Kura-Araxes spread, but a key factor in enabling it. The ability to adjust herd management, settlement rhythms, and household organization allowed communities to adapt to local conditions while participating in wider regional networks. Rather than being driven solely by push-pull dynamics or large-scale migration [267], the spread of the Kura-Araxes horizon across Southwest Asia may have rested on a foundational capacity for modular adaptation. We suggest that the Kura-Araxes horizon is best understood not as a singular, static culture, but as a constellation of seasonally organised, socially flexible, and ecologically responsive communities. Shared material culture may have provided a symbolic thread, but beneath this coherence lay a diversity of locally grounded practices. Maxta I offers a clear example of this pattern, a valuable case study for rethinking how cultural traditions expanded in the Early Bronze Age, not through the diffusion of a uniform way of life, but via the emergence of dynamic, interconnected networks shaped by social, economic, and pastoral adaptability.

Our conclusions are drawn from a single site and should be seen as a contribution to an ongoing regional debate rather than a comprehensive explanation of Kura-Araxes lifeways. While the Maxta I evidence challenges some long-standing assumptions about mobility and settlement, it does not resolve the full variability of pastoral strategies across the Kura-Araxes world and will require testing against additional multi-proxy datasets from other sites.

**Limitations of the study**

Although Maxta I provides an exceptional case study for multi-proxy reconstruction of Kura-Araxes pastoral systems, several limitations affect the resolution and scope of the findings. The ZooMS analysis, while successful, was based on a limited sample size of only 20 specimens, which constrains the taxonomic breadth and likely does not capture the full range of livestock species present at the site. Reconstructions of mobility and seasonal management relied exclusively on sequential stable carbon and oxygen isotope data, alongside dental wear analysis. While these techniques offer valuable insights into vertical transhumance and herd seasonality, the addition of strontium isotope analysis would be necessary to determine the specific geographic extent of mobility. Future work could also apply incremental growth band analysis of tooth cementum [268] to determine the season of slaughter and thereby provide an independent test of the seasonal herding model proposed here. In the absence of a full multi-proxy comparison of lowland and highland pasture/fodder systems, interpretations of herding mobility at Maxta I based on isotopic patterns should be regarded as a working hypothesis. The small number of specimens analysed (n = 10) also underscores the biographical nature of the data, reflecting the life histories of a few individuals rather than population-wide trends. Similarly, birth seasonality models were based on clear sinusoidal isotope sequences from only six individuals; the exclusion of non-modelable sequences may introduce bias, particularly if certain birth timings are underrepresented. Moreover, modern reference datasets used to estimate birth timing are primarily based on European caprine populations, which may not accurately reflect the reproductive ecology of ancient herds in the South Caucasus.

A key limitation of this study is the absence of modern isotopic data for $C_3$ and $C_4$ plants from the immediate surroundings of Maxta. As a result, our comparative thresholds rely on published regional datasets. These may not capture the full local isotopic variability of plant communities. The values used here should therefore be regarded as approximate regional references rather than precise local baselines, and further sampling of modern vegetation in Nakhchivan is needed to refine these estimates.

Although the present study focuses on sheep and goats, cattle constituted a substantial part of the reconstructed herd and likely contributed to secondary products such as milk, dung (used as fuel and manure), traction, and meat. However, current evidence for cattle is restricted to species counts, with limited ageing and no isotopic data available to reconstruct their seasonal management. Future analyses will be essential to clarify the role of cattle within the seasonal agro-pastoral system proposed here. Time constraints during fieldwork further restricted the collection of metric data, reducing the ability to assess demographic profiles or to confidently distinguish between wild and domestic taxa.

## Supporting information

**S1 File. Environmental Background.**
(DOCX)

**S2 File. Additional Images.**
(DOCX)

**S3 Data. Caucasus Oxygen isotopes.**
(XLSX)

**S4 Data. Caucasus Vegetation.**
(XLSX)

**S5 Data. Zooarchaeology.**
(XLSX)

**S6 Data. Stable Isotope Analysis.**
(XLSX)

**S7 Data. Seasonality of Birth.**
(XLSX)

**S8 Data. Phaseshift modelling.**
(XLSX)

**S9 File. Images of Sequentially Sampled Teeth.**
(PDF)

## Author contributions

**Conceptualization:** Gwendoline Maurer, Narmin Ismayilova.

**Data curation:** Gwendoline Maurer.

**Formal analysis:** Gwendoline Maurer, Delphine Frémondeau, Alexandra Nederbragt, Anne-Lise Jourdan, Rachel Wood, Rhiannon E. Stevens.

**Funding acquisition:** Gwendoline Maurer, Narmin Ismayilova, Safar Ashurov, Veli Bakhshaliyev.

**Investigation:** Gwendoline Maurer, Delphine Frémondeau, Narmin Ismayilova, Safar Ashurov, Fidan Khalafova Aliyeva, Alexandra Nederbragt, Anne-Lise Jourdan, Rachel Wood.

**Methodology:** Gwendoline Maurer, Delphine Frémondeau, Narmin Ismayilova, Alexandra Nederbragt, Anne-Lise Jourdan, Rachel Wood, Rhiannon E. Stevens.

**Project administration:** Gwendoline Maurer, Safar Ashurov, Veli Bakhshaliyev.

**Resources:** Safar Ashurov, Veli Bakhshaliyev, Rachel Wood, Louise Martin, Rhiannon E. Stevens.

**Software:** Gwendoline Maurer.

**Supervision:** Safar Ashurov, David Wengrow, Louise Martin, Rhiannon E. Stevens.

**Validation:** Gwendoline Maurer, Narmin Ismayilova, Safar Ashurov, Rachel Wood, Louise Martin, Rhiannon E. Stevens.

**Visualization:** Gwendoline Maurer, Safar Ashurov, Fidan Khalafova Aliyeva.

**Writing – original draft:** Gwendoline Maurer, Delphine Frémondeau, Narmin Ismayilova, Safar Ashurov, Alexandra Nederbragt, Anne-Lise Jourdan, David Wengrow, Rachel Wood, Louise Martin, Rhiannon E. Stevens.

**Writing – review & editing:** Gwendoline Maurer, Delphine Frémondeau, Narmin Ismayilova, Safar Ashurov, Veli Bakhshaliyev, Fidan Khalafova Aliyeva, Alexandra Nederbragt, Anne-Lise Jourdan, David Wengrow, Rachel Wood, Louise Martin, Rhiannon E. Stevens.

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
