## [Decision Letter · Decision Letter 0]

18 Sep 2025

Dear Dr. Maurer,

Thank you for submitting your manuscript to PLOS ONE. After careful consideration, we feel that it has merit but does not fully meet PLOS ONE’s publication criteria as it currently stands. Therefore, we invite you to submit a revised version of the manuscript that addresses the points raised during the review process.

Your manuscript has now been seen by two referees, whose comments are appended below. You will see from these comments that while the referees find your work of potential interest, they have raised substantial concerns that must be addressed. In light of these comments, we cannot accept the manuscript for publication, but would be interested in considering a revised version that addresses these serious concerns.

We hope you will find the referees' comments useful as you decide how to proceed. Should presentation of further data and analysis allow you to address these criticisms, we would be happy to look at a substantially revised manuscript. However, please bear in mind that we will be reluctant to approach the referees again in the absence of major revisions.

We look forward to receiving your revised manuscript.

Kind regards,

Peter F. Biehl, PhD

Academic Editor

PLOS ONE

3. In your manuscript, please provide additional information regarding the specimens used in your study. Ensure that you have reported human remain specimen numbers and complete repository information, including museum name and geographic location.

For more information on PLOS One's requirements for paleontology and archeology research, see https://journals.plos.org/plosone/s/submission-guidelines#loc-paleontology-and-archaeology-research.

“This research was funded by the London Arts & Humanities Partnership (LAHP), the UCL Institute of Archaeology, the Ministry of Education of the Republic of Azerbaijan and the National Environmental Isotope Facility (NEIF Grant 2777).”

“This research was funded by the London Arts & Humanities Partnership (LAHP) (Authors: GM), the UCL Institute of Archaeology Small Research Grant (Authors:GM), the Ministry of Education of the Republic of Azerbaijan (AUthors: NI) and the National Environmental Isotope Facility (NEIF Grant 2777) (Authors: GM & NI)

https://www.lahp.ac.uk/

https://edu.gov.az/en/

https://www.isotopesuk.org/

https://www.ucl.ac.uk/archaeology/

The sponsors or funders did not play any role in the study design, data collection and analysis, decision to publish, or preparation of the manuscript”

6. We note that you have referenced **(Batiuk, S. Migration Theory and the Distribution of the Early Transcaucasian Culture. Unpublished PhD dissertation, University of Toronto (2005**) and ) and ) and ) and **(Longford, C. Plant Economy of the Kura-Araxes: A Comparative Analysis of Agriculture in the Near East from the Chalcolithic to the Middle Bronze Age. Vol. 1 & 2. Unpublished PhD thesis, University of Sheffield, Sheffield (2015))** which has currently not yet been accepted for publication. Please remove this from your References and amend this to state in the body of your manuscript: (ie “Bewick et al. [Unpublished]”) as detailed online in our guide for authorswhich has currently not yet been accepted for publication. Please remove this from your References and amend this to state in the body of your manuscript: (ie “Bewick et al. [Unpublished]”) as detailed online in our guide for authorswhich has currently not yet been accepted for publication. Please remove this from your References and amend this to state in the body of your manuscript: (ie “Bewick et al. [Unpublished]”) as detailed online in our guide for authorswhich has currently not yet been accepted for publication. Please remove this from your References and amend this to state in the body of your manuscript: (ie “Bewick et al. [Unpublished]”) as detailed online in our guide for authors

7. We note that Figures 1, 3 and 4 in your submission contain [map/satellite] images which may be copyrighted. All PLOS content is published under the Creative Commons Attribution License (CC BY 4.0), which means that the manuscript, images, and Supporting Information files will be freely available online, and any third party is permitted to access, download, copy, distribute, and use these materials in any way, even commercially, with proper attribution. For these reasons, we cannot publish previously copyrighted maps or satellite images created using proprietary data, such as Google software (Google Maps, Street View, and Earth). For more information, see our copyright guidelines: http://journals.plos.org/plosone/s/licenses-and-copyright.

1. You may seek permission from the original copyright holder of Figures 1, 3 and 4 to publish the content specifically under the CC BY 4.0 license.

8. We note that there is identifying data in the Supporting Information file < S5_Selected samples isotope analysis.xlsx and S7_collagen isotope analysis.xlsx >. Due to the inclusion of these potentially identifying data, we have removed this file from your file inventory. Prior to sharing human research participant data, authors should consult with an ethics committee to ensure data are shared in accordance with participant consent and all applicable local laws.

-Location data

Please remove or anonymize all personal information, ensure that the data shared are in accordance with participant consent, and re-upload a fully anonymized data set. Please note that spreadsheet columns with personal information must be removed and not hidden as all hidden columns will appear in the published file.

Additional Editor Comments:

Your manuscript has now been seen by two referees, whose comments are appended below. You will see from these comments that while the referees find your work of potential interest, they have raised substantial concerns that must be addressed. In light of these comments, we cannot accept the manuscript for publication, but would be interested in considering a revised version that addresses these serious concerns.

We hope you will find the referees' comments useful as you decide how to proceed. Should presentation of further data and analysis allow you to address these criticisms, we would be happy to look at a substantially revised manuscript. However, please bear in mind that we will be reluctant to approach the referees again in the absence of major revisions.

Reviewers' comments:

Reviewer's Responses to Questions

**Comments to the Author**

1. Is the manuscript technically sound, and do the data support the conclusions?

Reviewer #1: Yes

Reviewer #2: Partly

2. Has the statistical analysis been performed appropriately and rigorously?

Reviewer #1: Yes

Reviewer #2: I Don't Know

3. Have the authors made all data underlying the findings in their manuscript fully available?

Reviewer #1: Yes

Reviewer #2: Yes

4. Is the manuscript presented in an intelligible fashion and written in standard English?

Reviewer #1: Yes

Reviewer #2: Yes

Reviewer #1: This manuscript provides a comprehensive analysis of seasonal pastoral strategies employed by the population of Maxta I ascribed to the Kura-Araxes culture in Nakhchivan, Azerbaijan, southern Caucasus. The proposed multi-method approach included an extensive program of studies based on a variety of methods employed to identify seasonal mobility of the herders related to development of upland herding. The comparison of pastoral strategies noted for their diversity provides an opportunity to discuss economic activity, temporal and spatial organization of the Maxta I occupants’ settlement strategy, assess the level of adaption to local landscapes, exploited resources as well as seasonal mobility within the exploited lowland–upland pastures.

Based on the case study of a lowland settlement, the authors use the seasonality/mobility model they have developed to reconstruct one of the most important components of the Kura-Araxes subsistence model based on the agro-pastoral system, provide evidence supporting use of diverse strategies in a rather large region occupied by this culture across southwest Asia and suggest that this economic potential also existed in other Kura-Araxes areas. This model is based on livestock management when a greater part of caprines is seasonally moved between the upland and lowland pastures while a smaller part is raised at the site or nearby. The authors note dominance of sheep in the faunal assemblage compared to goats, almost complete absence of suids, and a rather high percentage of wild animal bones. This paper is the first attempt to analyze this dual economic model for the Kura-Araxes culture when some residents remained at the settlement year-round engaging in various household activities while other occupants moved herds to nearby and distant pastures.

There is no doubt that this study is exciting as it shows advantages of comprehensive approaches in the analyses of archaezoological material making it possible to discuss economic strategies in prehistory. The paper can be recommended for publication in PLOS ONE. However, it needs some clarifications that will reinforce main conclusions derived by the authors.

1. It is better to provide the results of 14C dating available for the site structures analyzed, at least in the Suppl. Data. Figure 2 that the authors make reference to (pg. 5) does not contain any dating results.

2. The authors should provide stronger arguments in support of such sector as wool production. The authors use the estimated mortality rates of animals (sheep and goats) as the main evidence of wool production and say that the animal slaughter age can point to such secondary products as milk and wool. Examples of using both plant (made from flax) and wool textiles in the southern Caucasus, at least, in contemporaneous cultures where such textiles have been identified should be provided (see in Ananauri big kurgan № 3. Tbilisi: Georgian National Museum/ 2016; Bedianashvili, G., Jamieson, A., Longford, C., Martkoplishvili, I., Paul, J., Sagona, C., 2022. Evidence for textile production in Rabati, Georgia, during the Bedeni phase of the Early Kurgan period. Journal of Archaeological Sciences: Reports 43, 103467. ISSN 2352-409X). More importantly, the recent studies of wool textile from the southern Caucasus have shown that it is made of goat underwool (Archaeological Research in Asia 43 (2025) 100635). The results of the study are well correlated with the data from Arslan-Tepe.

Using the results of tracewear analysis of similar items from other Bronze Age sites, and judging by the form of the horse ulnar bones presented in Fig. 2 (c) and in the Supplementary file, these ulnar bones could not be used in textile production. Most likely, these items are piercers or perforators similar to tools used in leather production that, presumably, was one of main productions at the settlement.

It would be interesting to know if there are textile impressions on clay vessels or coating on the dwellings.

3. The taphonomic analyses of animal bones are not very clear. Are there cases when the anatomical bone composition of a specific animal species includes all elements of the skeleton? If there are, it may suggest that animal carcasses were cut into pieces at the site. First of all, it holds true for cattle. Presence of bones of newly born animals of these species may also indicate that cattle was not only kept but also raised at the site. Please clarify if such data are available.

4. Probably, the authors of this study did some experimenting cut of grass in summer pastures (plots) located around the site to determine percentages of C3 and C4 plants in such pastures. Such data, if available, could reinforce the suggestions made by the authors regarding use of summer pastures with pure C3 vegetation located far from the site.

Maybe, it will be useful to indicate what trees and shrubs grow around the site as the authors make reference to browsers which fed on leaves and shoots in winter. Such data, certainly, cannot be used as direct evidence of the landscape in prehistory. However, the authors argue that the climate at the time of Maxta I was similar to modern climate; therefore, this information can help get an insight in core fodder resources in winter at the time of the site occupation.

5. Probably, it should be indicated that the snowfree period (according to modern climatic data with which the authors correlate presumed local climatic characteristics during the period when the site was occupied) lasted almost 11 months while the average snow cover does not exceed 25 mm; for this reason, practically all local pastures around the settlement can be used during the cold time of the year.

6. It is not quite clear if there are differences in the nitrogen and carbon isotopic composition of the animals that grazed on the manured pastures around the settlement and those that were moved to more extensive upland pastures, the data on the δ15N values of the analyzed animal sample are similar. And while the authors note that there is no direct evidence of haymaking, was the use of straw possible? For example, are there any impressions of straw on the walls and the floor of the mud-brick and wattle-and-daub dwellings?

7. Reference to the archaeobotanical data obtained from the settlement integrated into the analyses (line 648) is absent, though it is made below (line 763). As the authors propose the seasonal adaptive agro-pastoral model, the available archaeobotanical data (wheat, barley, fruits as indicated in the model in Figure 8) should be provided in the text. Have any straw impressions been documented on the mud-brick and wattle-and-daub dwellings? Such data can imply that winter fodder (straw) was hoarded.

8. The seasonality of the settlement is based on the modelling of δ18O sequences and variations in the δ13С values in the intra-tooth samples, seasonality of birth, absence of newly born caprines among the settlement animals (which means lambing/kidding outside the settlement in the spring and autumn for sheep and late autumn for goats), as well as variations in the collagen stable δ13С and δ15N values in the animal bones.

According to the authors, they plan to analyze variations in the strontium isotopic ratios which will confirm one of the conclusions on lambing and kidding outside the settlement in the summer and autumn pastures.

The authors should also note that there is one more method that can be used to confirm or reject the conclusions regarding year-round or seasonal grazing of various species of animals in the pastures, it is the dentum analysis of incremental bands in the cementum and detrine of teeth (Klevezal G.A. Recording structures of mammals. Determination of age and reconstruction of life history. Rotterdam: Balkema Publish. House. 1996). This method is employed to determine the season of animal slaughter and thereby confirm the seasonal pastoral model under discussion.

9. The paper discussed focuses on the analysis of sheep/goats; however, the cattle is the second largest share of the reconstructed herd of the local population. Apparently, cattle played an important role in secondary productions, especially, milking, dung collection (used as fuel and fertilizer for fields), traction as well as meat production. The role of such animals in the discussed seasonal agro-pastoral model should be assessed, e.g. was all cattle raised at the site? Was the cattle kept in the stalls in winter and did it graze on pastures around the settlement in summer, how far could such pastures be located?

10. The authors should carefully check numbering of the references in the text and the references list. See, for example, line 795: In the References section Frachetti paper is with the number 199, but in the text it is numbered 200.

Reviewer #2: This paper mainly presents archaeozoological and isotopic results that helps describing the agro-pastoral strategies of the Kura-Araxes inhabitants of Maxta.

The main problem of the paper is the over interpretation of the data. And also, the tendency to inflate the impact of the study is clumsy. It is a decent study but it is not establishing a strong model to understand the diversity of pastoral strategies in SW Asia. Azerbaijan is a small territory and Nakhchivan even more. It is home to a dozen of archeological excavations but of course it is necessary to see the image from the whole region as there are interesting projects in Armenia and Georgia as well. Therefore a sentence such as “Maxta I is among the few systematically excavated Kura-Araxes Culture (KAC) sites in Nakhchivan, and the only one in Azerbaijan where pastoral mobility and economic organisation are investigated in depth” sounds clumsy.

The paper should be refocused on the produced data and their interpretation. Part of the text and the supplementary data are not fully relevant to the aim of the paper. For these reasons I recommend a major revision.

There is an impressive list of 282 references. It’s too much. It is not necessary to show that you have read everything. This is not a review article. It is necessary to be more synthetic. Furthermore, in the introduction to the Kura-Araxes culture, there is no hierarchy in the sources. Everything is cited from recognized syntheses to anecdotical papers. Some interesting references are however missing. You don’t mention the fierce debate about the origins of the Kura-Araxes that concerns evidence from Nakhchivan (Marro et al. (2014) On the Genesis of the Kura-Araxes Phenomenon: New Evidence from Nakhchivan (Azerbaijan), with reply from Palumbi and Chataigner, and reply of Marro et al). You do not either discuss the place or specificities of Maxta in comparison to other KA settlements excavated in Nakhchivan or in the region.

Line 139. I don’t see the link between Figure 2 and the fact that bones from Maxta are currently radiocarbon dated.

Figure 3 could be completed with the hydrological system (main rivers and lakes), the name of the countries and maybe a

few modern cities to help the reader.

Figure 4. Is not very useful in the text but called many times in the supplementary document, it should be moved there.

When you mention modern δ18O values, in the supplementary for instance, you use modeled values (from Bowen & Revenaugh) but the way you present them isn’t clear, it looks like you refer to measured values, it is different. Line 177 for example. Aren't there actual values from the GNIP in the area ?

Line 186. many publications report paleoenvironmental data from the Caucasus to investigate the climatic and environmental conditions in the past. References 105 and 106 are not relevant.

Line 233. It would be interesting to know what the sub-set represents in % of the whole assemblage. How the subset was selected?

Line 287-290. I think that it would be worth to move supplementary 8 into the main text. It is very relevant for the discussion

Line 435-47. The collected data are not relevant due to the variability of the sampling location, and the number of values for C3 plants, to build a regional threshold. You should first explain why the common thresholds used in previous articles reporting isotopic data, needed to be redefined, if this is really the case. Line 446 you mention -4.8‰ for pure C4 but 4.8 in the supplementary data.

ZooMs. It is difficult to understand figure 5e. You mention in the text 20 samples from ZooMs but in the figure it is written N=26

Lines 600-619. I do not agree with the idea that pattern A.3 correspond to animals grazing in highlands in the summer. They display carbon values around -9‰. It is not low enough for the highlands. Calculation by Samei (2013) Highland pastoralism in the Early Bronze Age Kura-Araxes cultural tradition: "Stable oxygen (δ^18^O) and carbon (δO) and carbon (δO) and carbon (δO) and carbon (δ^13^C) isotope analyses of herd mobility at Köhne Shahar in northwestern Iran", and also modern transhumant sheep from Armenia (published in Janzen et al 2023 Neolithic herding practices in the Southern Caucasus: Animal management in the early 6C) isotope analyses of herd mobility at Köhne Shahar in northwestern Iran", and also modern transhumant sheep from Armenia (published in Janzen et al 2023 Neolithic herding practices in the Southern Caucasus: Animal management in the early 6C) isotope analyses of herd mobility at Köhne Shahar in northwestern Iran", and also modern transhumant sheep from Armenia (published in Janzen et al 2023 Neolithic herding practices in the Southern Caucasus: Animal management in the early 6C) isotope analyses of herd mobility at Köhne Shahar in northwestern Iran", and also modern transhumant sheep from Armenia (published in Janzen et al 2023 Neolithic herding practices in the Southern Caucasus: Animal management in the early 6^th^ millennium BCE at Masis Blur in Armenia’s Ararat Valley)" show that alpine meadows correspond to values around -12‰ in the enamel. The anti-phase relationship is due to a C4 enriched diet in the winter, not to highland pastures in the summer. Your main evidence is that Maxta is in an arid environment full of C4 plants. Actually, the landscape around Maxta is not arid, as it can be seen from a google earth image. The valley of the Araxes is rather marshy and it was probably even more in the past before the control of the Araxes river by dams. Arid and salty environments can be found in the area, probably more on the hilly flanks, but considering that Maxta should have a C4 signal for local grazing might be wrongmillennium BCE at Masis Blur in Armenia’s Ararat Valley)" show that alpine meadows correspond to values around -12‰ in the enamel. The anti-phase relationship is due to a C4 enriched diet in the winter, not to highland pastures in the summer. Your main evidence is that Maxta is in an arid environment full of C4 plants. Actually, the landscape around Maxta is not arid, as it can be seen from a google earth image. The valley of the Araxes is rather marshy and it was probably even more in the past before the control of the Araxes river by dams. Arid and salty environments can be found in the area, probably more on the hilly flanks, but considering that Maxta should have a C4 signal for local grazing might be wrongmillennium BCE at Masis Blur in Armenia’s Ararat Valley)" show that alpine meadows correspond to values around -12‰ in the enamel. The anti-phase relationship is due to a C4 enriched diet in the winter, not to highland pastures in the summer. Your main evidence is that Maxta is in an arid environment full of C4 plants. Actually, the landscape around Maxta is not arid, as it can be seen from a google earth image. The valley of the Araxes is rather marshy and it was probably even more in the past before the control of the Araxes river by dams. Arid and salty environments can be found in the area, probably more on the hilly flanks, but considering that Maxta should have a C4 signal for local grazing might be wrongmillennium BCE at Masis Blur in Armenia’s Ararat Valley)" show that alpine meadows correspond to values around -12‰ in the enamel. The anti-phase relationship is due to a C4 enriched diet in the winter, not to highland pastures in the summer. Your main evidence is that Maxta is in an arid environment full of C4 plants. Actually, the landscape around Maxta is not arid, as it can be seen from a google earth image. The valley of the Araxes is rather marshy and it was probably even more in the past before the control of the Araxes river by dams. Arid and salty environments can be found in the area, probably more on the hilly flanks, but considering that Maxta should have a C4 signal for local grazing might be wrong

Discussion. Line 653-654 you mention that part of the flock remained year-round at the site. However, lines 613-614 your interpretation of the isotopic data suggest the contrary.

Figure 8 is quite problematic because it is a theoretical model that is not fully supported by your data. Its inclusion in the text gives the impression that it is the model for Kura-Araxes Maxta. Autumn birth and winter lactation are not proven. You have one individual out of five born in the summer. It could be an out-of-season born individual that survived. You don’t have enough data to suggest that there was an attempt for the herders to have two distinct birth seasons in order to have two lactation seasons. I don’t even speak about the goat as there are not enough reference data to precisely attribute Xo/X values to a birth season.

Mountain pasture in the summer is not supported by enough evidence (see my comment above about the 13C values expected from highland pastures).

Sheep and goat flocks could have been used to manure the field. However, to evaluate the real impact of manuring, it would necessitate to have isotopic values from botanical remains from Maxta. To discuss the seasonality of sowing, again a detailed botanical study is needed.

The archaeobotanical evidence you mention line 763 are from the 2x2 sounding in 2006, how representative are they of the whole KA agriculture at Maxta?

The limitations of the study are clearly described at the end of the article. However it is not enough to justify the discussion that is clearly aimed at justifying a theoretical model that doesn’t completely fit with the reality of the data.

The supplementary file “environmental background” also includes photographs of bone tools, ceramics, archaeological structures and zoomorphic figures. It is not clear to me why it is in the same document.

Supplementary 2. Caucasus vegetation. It is not clear to me in which way the list of C4 and C3 plant species support the discussion of the paper. Concerning the δ13C values of C3 and C4 plants, the values reported from Rudov et al were obtained from plant from all over SW Asia. Is not necessarily helpful to decide whether Maxta animals consumed or not C4 plants. The values published by Shishlina and al are mostly coming from the Russian steppes, very far away from the Araxes basin and ecologically very different.

Supplementary 5. Would be nice to add species and side.

Supplementary 9. I would be necessary to explain the abbreviations and the calculations

.

Reviewer #1: No

Reviewer #2: No

---

## [Author Response · Author response to Decision Letter 1]

3 Nov 2025

We thank the Academic Editor and the reviewers for their detailed and constructive comments, which have greatly improved our manuscript. Below we provide a detailed response to each point. Reviewer/Editor comments are shown in black, followed by our responses in blue.

We have formatted the manuscript according to the PLOS One guidelines.

2. Please include a complete copy of PLOS’ questionnaire on inclusivity in global research in your revised manuscript. Our policy for research in this area aims to improve transparency in the reporting of research performed outside of researchers’ own country or community. The policy applies to researchers who have travelled to a different country to conduct research, research with Indigenous populations or their lands, and research on cultural artefacts. The questionnaire can also be requested at the journal’s discretion for any other submissions, even if these conditions are not met. Please find more information on the policy and a link to download a blank copy of the questionnaire here: Best Practices in Research Reporting | PLOS One. Please upload a completed version of your questionnaire as Supporting Information when you resubmit your manuscript.

We have completed and added the questionnaire.

3. In your manuscript, please provide additional information regarding the specimens used in your study. Ensure that you have reported human remain specimen numbers and complete repository information, including museum name and geographic location.

For more information on PLOS One's requirements for paleontology and archeology research, see https://journals.plos.org/plosone/s/submission-guidelines#loc-paleontology-and-archaeology-research.

We have added to the Data and Code availability section the following statement: [page 45]

All necessary permits were obtained for the described study, which complied with all relevant regulations.

“This research was funded by the London Arts & Humanities Partnership (LAHP), the UCL Institute of Archaeology, the Ministry of Education of the Republic of Azerbaijan and the National Environmental Isotope Facility (NEIF Grant 2777).”

“This research was funded by the London Arts & Humanities Partnership (LAHP) (Authors: GM), the UCL Institute of Archaeology Small Research Grant (Authors: GM), the Ministry of Education of the Republic of Azerbaijan (Authors: NI) and the National Environmental Isotope Facility (NEIF Grant 2777) (Authors: GM & NI)

https://www.lahp.ac.uk/

https://edu.gov.az/en/

https://www.isotopesuk.org/

https://www.ucl.ac.uk/archaeology/

The sponsors or funders did not play any role in the study design, data collection and analysis, decision to publish, or preparation of the manuscript”

Thank you for letting us know. We have added the amended statement to the cover letter. We have now removed it from the manuscript main text.

“This research was funded by the London Arts & Humanities Partnership (LAHP) (Authors: GM), the UCL Institute of Archaeology Small Research Grant (Authors: GM), the Ministry of Education of the Republic of Azerbaijan (Authors: NI) and the National Environmental Isotope Facility (NEIF Grant 2777) (Authors: GM & NI)

https://www.lahp.ac.uk/

https://edu.gov.az/en/

https://www.isotopesuk.org/

https://www.ucl.ac.uk/archaeology/

The sponsors or funders did not play any role in the study design, data collection and analysis, decision to publish, or preparation of the manuscript”

The data is now accessible on Mendeley: https://data.mendeley.com/preview/24dyjf7c5j?a=30f08d2e-3750-4742-bf01-3c54f3cd36a6

6. We note that you have referenced (Batiuk, S. Migration Theory and the Distribution of the Early Transcaucasian Culture. Unpublished PhD dissertation, University of Toronto (2005) and (Longford, C. Plant Economy of the Kura-Araxes: A Comparative Analysis of Agriculture in the Near East from the Chalcolithic to the Middle Bronze Age. Vol. 1 & 2. Unpublished PhD thesis, University of Sheffield, Sheffield (2015)) which has currently not yet been accepted for publication. Please remove this from your References and amend this to state in the body of your manuscript: (ie “Bewick et al. [Unpublished]”) as detailed online in our guide for authors

They have both been removed from the in text references and bibliography.

7. We note that Figures 1, 3 and 4 in your submission contain [map/satellite] images which may be copyrighted. All PLOS content is published under the Creative Commons Attribution License (CC BY 4.0), which means that the manuscript, images, and Supporting Information files will be freely available online, and any third party is permitted to access, download, copy, distribute, and use these materials in any way, even commercially, with proper attribution. For these reasons, we cannot publish previously copyrighted maps or satellite images created using proprietary data, such as Google software (Google Maps, Street View, and Earth). For more information, see our copyright guidelines: http://journals.plos.org/plosone/s/licenses-and-copyright.

1. You may seek permission from the original copyright holder of Figures 1, 3 and 4 to publish the content specifically under the CC BY 4.0 license.

Thank you for your message regarding Figures 1, 3, and 4.

We have reviewed all figures to ensure full compliance with the CC BY 4.0 license.

Figure 3 (now Figure 1): Location and elevation map of Maxta I, created using data from the ASTER Global Digital Elevation Model (GDEM) Version 3, a public NASA/METI product available via NASA Earthdata. Data processed in ArcMap v10.8.2.

Figure 4 (now Figure 2): Spatial variation of temperature, precipitation, and snow cover across the Caucasus, based on open-access NASA datasets from Giovanni, NASA GES DISC, processed in ArcMap v10.8.2 and assembled using BioRender.com.

Both datasets are distributed under NASA’s Open Data and Information Policy, which permits unrestricted reuse with attribution.

To comply fully with copyright requirements, we have removed Figure 1 (originally based on an Esri basemap).

8. We note that there is identifying data in the Supporting Information file < S5_Selected samples isotope analysis.xlsx and S7_collagen isotope analysis.xlsx >. Due to the inclusion of these potentially identifying data, we have removed this file from your file inventory. Prior to sharing human research participant data, authors should consult with an ethics committee to ensure data are shared in accordance with participant consent and all applicable local laws.

-Location data

Please remove or anonymize all personal information, ensure that the data shared are in accordance with participant consent, and re-upload a fully anonymized data set. Please note that spreadsheet columns with personal information must be removed and not hidden as all hidden columns will appear in the published file.

Thank you. This is data from archaeological sheep and goat specimens not from humans.

We have now added captions for our Supporting Information files at the end of the manuscript as well as updated any in-text citations.

Reviewers' comments:

Reviewer #1:

This manuscript provides a comprehensive analysis of seasonal pastoral strategies employed by the population of Maxta I ascribed to the Kura-Araxes culture in Nakhchivan, Azerbaijan, southern Caucasus. The proposed multi-method approach included an extensive program of studies based on a variety of methods employed to identify seasonal mobility of the herders related to development of upland herding. The comparison of pastoral strategies noted for their diversity provides an opportunity to discuss economic activity, temporal and spatial organization of the Maxta I occupants’ settlement strategy, assess the level of adaptation to local landscapes, exploited resources as well as seasonal mobility within the exploited lowland–upland pastures.

Based on the case study of a lowland settlement, the authors use the seasonality/mobility model they have developed to reconstruct one of the most important components of the Kura-Araxes subsistence model based on the agro-pastoral system, provide evidence supporting use of diverse strategies in a rather large region occupied by this culture across southwest Asia and suggest that this economic potential also existed in other Kura-Araxes areas. This model is based on livestock management when a greater part of caprines is seasonally moved between the upland and lowland pastures while a smaller part is raised at the site or nearby. The authors note dominance of sheep in the faunal assemblage compared to goats, almost complete absence of suids, and a rather high percentage of wild animal bones. This paper is the first attempt to analyze this dual economic model for the Kura-Araxes culture when some residents remained at the settlement year-round engaging in various household activities while other occupants moved herds to nearby and distant pastures.

There is no doubt that this study is exciting as it shows advantages of comprehensive approaches in the analyses of archaezoological material making it possible to discuss economic strategies in prehistory. The paper can be recommended for publication in PLOS ONE. However, it needs some clarifications that will reinforce main conclusions derived by the authors.

We thank the r

---

## [Decision Letter · Decision Letter 1]

26 Dec 2025

Dear Dr. Maurer,

Thank you for submitting your manuscript to PLOS ONE. After careful consideration, we feel that it has merit but does not fully meet PLOS ONE’s publication criteria as it currently stands. Therefore, we invite you to submit a revised version of the manuscript that addresses the points raised during the review process.

We look forward to receiving your revised manuscript.

Kind regards,

Peter F. Biehl, PhD

Academic Editor

PLOS One

Journal Requirements:

Additional Editor Comments:

Please address all the remaining changes in detail before re-submission.

Reviewers' comments:

Reviewer's Responses to Questions

**Comments to the Author**

Reviewer #1: All comments have been addressed

Reviewer #2: (No Response)

2. Is the manuscript technically sound, and do the data support the conclusions?

Reviewer #1: Yes

Reviewer #2: Partly

3. Has the statistical analysis been performed appropriately and rigorously?

Reviewer #1: Yes

Reviewer #2: N/A

4. Have the authors made all data underlying the findings in their manuscript fully available?

Reviewer #1: Yes

Reviewer #2: Yes

5. Is the manuscript presented in an intelligible fashion and written in standard English?

Reviewer #1: Yes

Reviewer #2: Yes

Reviewer #1: The second variant of the manuscript includes many changes, the outline of the paper presentation has been changed and some other changes have been incorporated. On the whole, the authors of the paper have provided explanation and clarification to most of the questions. The presented overall economic model used by one specific population group that lived at the Maxtra I settlement is based on development of pasture sheep/goat raising and cattle raising near the settlement; this population also cultivated crops. The revised text includes data of paleoarchaebotany. Generally speaking, this model is consistent with the economic model of the Caucasus inhabitants throughout the Middle Ages and the early 20th century where raising of ovicaprids played a key role.

The pending issue is to what extent raising of ovicaprids and use of adjacent pastures were correlated with the area of cultivated lands. It is clear that the author focused, mainly, on the primary role of extensive sheep/goat raising and related secondary productions; therefore, the geochemical data on the archaeozoological samples reflect a mobile nature of animal husbandry. However, the paper does not contain any multi-proxy analysis of the pasture/fodder system near the settlement and the system in the foothill area or their comparison; for this reason, the proposed interpretation remains a working hypothesis.

In the absence of the data on a significant role played by secondary products, first of all, data on the production of wool raw material, woolen cloths, and felt, I think that sheep and goats were raised to obtain meat, and, possibly, milk on a seasonal basis. I would like to note that ethnographic and modern data suggest that the lactation period of sheep in the Caucasus, probably, lasted for not more than two months or up to three months in some areas.

The issue of localization of winter pastures and fodder storage for winter has not yet been addressed. The authors analyze a potential environmental situation similar to the modern situation and think that the population could feed sheep/goats as well as cattle on pastures almost during the entire winter. The steppe landscape around Maxtra-1 seems to be consistent with this pasture strategy whereas foothill and mountainous pastures could be used during the warm period of the year. On the whole, while the authors mention that the steppe landscape and natural zone determined the type of the economic system developed by the Maxtra-I population, this statement should be spelled out with more clarity. Besides, the ethnography of the Caucasus highlanders suggests that shepherds often moved with their sheep from one area of mountainous summer pastures to another. Usually such movements lasted for 5-6 days because of overgrazing and time needed to restore pasture cover. This type of mobility should also be taken into account in developing the economic model; it is also important to take into account that, based on the ethnography of the Caucasus, only men of different ages were involved in such seasonal moves.

The proposed agro-pastoral model is based on new archaeozoological, ZooMS and isotope data, obtained for one site of the Kuro-Araxes culture and so far it remains to be the only model of this type. This model is the main outcome of the study and the basis for follow-up comparative analysis with other models that will be developed using materials from other sites. The paper is recommended for publication with minor changes and clarifications.

Minor comments

Line 76 – more precise chronology of the Majkop culture of the northern Caucasus is 3650-3000 calBC. The dating of the human samples from the steppe region affected by a reservoir effect shifted the lower boundary backward to 3900 calBC, making the date older.

It is better to move lines 181-188 (description of Maxtra I location) up and put it before line 140 as well as descriptions of the ….. given further in the text.

Reviewer #2: Thank you for the revised version of your manuscript. Although you answered to all my comments, some part of the discussion needs to be strengthened. I also have additional comments.

L. 163 you should consider using the azeri spelling Çaqqallıqtəpə because there is absolutely nothing in books or internet about Chaggalligtepe

l. 453-454. There is a shift of approximately 5-6 month between the beginning of the crown formation and the enamel maturation. Hence the isotopic record rather reflects the life of the individual between its 6 and 18 months (see references 211 - 212 -226)

l. 488-499. The proposed threshold for pure C3 diet is rather low (although the upper range of the threshold in enamel is -8.58 for d13C). It is rightly stated that Janzen et al proposed a threshold at -6.7 (and not quoted Messana et al. a threshold at -7.4 in southern Europe see doi 10.1007/s12520-024-02116-z). The difference with your threshold is not discussed enough although it might highly impact the interpretation of the herd's diet.

l. 552. If I understood well, 10% of the Bos primigenius bones are burnt. It means 1 bone (out of 10). On such frequency I would not comment the occurrence of burning. Same with the 9% of 11 large mammal remains. Considering that you studied approximately 25% of the assemblage, any significance of frequencies is difficult to assess.

Fig. 4. You write Taxonomic abundance in % of the NISP but the graph shows n=410 (which includes unidentified remains). And the caprines are shown at approx 55% while in the table in supp 5 they make 68% of the NISP.

L. 602-608. The text is not consistent with fig. 4 and Supplementary 6. The text and fig.4 mention 20 mandibles identified with Zoom (and 5 attributed to Ovis according to the morphology). In supp6, there are only 18 specimens. None is identified as sheep in the column F (morphology). And only 13 out of 18 are identified with Zoom (column G).

L. 619. You write "table Sx". Indeed the average values are missing from the supplementary data. It would be interesting for the reader to have these summary statistics. Boxplots to show the distribution of d13C and d18O in the different individuals would also be helpful.

L. 679-680. In variation A.2, the lowest d13C value is -10.3. You mention highland pastures. It should be discussed why -10.3 could correspond to highland pastures.

L. 703-705. In pattern 3, you assume pasture in high altitude because d13C values around -8.5 / -9 look to low for you to correspond to the Araxes valley. However, you used the same argument to identify high-altitude summer pasture in pattern 1 with d13C values around -10. It is not the same high-altitude pastures then?

In general, in the section “Sequential enamel bioapatite δ18O and δ13C values”, the values used to identify high altitude pasture are not sufficiently discussed. You don’t even mention your own article about Yeghegis-1 where values for this site at 1,500 masl are around -11 for d13C. Also, I’m not convinced by your main argument for vertical mobility when you consider that any values below the C3/C4 threshold rules out local pastures around Maxta I. The Sharur plain is one of the greenest spot of Nakhchivan, it is among the main hay provider areas of Nakhchivan. It is watered by both the Arpaçay river and the ground water of the Araxes. One can imagine green fields around the site (2-3 km radius). It is not proven that grazing on C4 plant was inevitable around Maxta I.

L. 710 About the section “seasonality of birth”. I checked the raw data you provided in S6 and S7. I think that it would be necessary to provide the pictures of the sampled teeth, so that the reader can see how the sampling was made and can see the ERJ.

For example, I have a question about the distance to erj of the samples of MI 18. There should be a gap of at least 1 mm between two samples. Because the procedure is to measure the sample position on the upper ridge (occlusal direction). The drill is 1mm thick. How can you have 0.5 or 0.6mm between samples?

I think there is also a mistake in modelling MI 25. In S7, x0 for this tooth is 6.5 mm. but in the graph it is clear that the max dO values are rather between 15 and 20. Maybe the model proposed a negative amplitude. If you correct, you will have a very different x0/X and birth season for this individual.

Models for MI 16, MI 18 and MI 42 are tentative because you cannot be sure that the first point of the sequence is really a minimum. However, you might argue, than even with a longer period X, the X0/X value would not be so different in term of birth season.

L. 726 Plotting the d15N and d13C in a graph would be easier for the reader. You could even use the graph you published in the Yeghegis-1 article to put these values into a broader context. Collagen values are not used enough in the discussion. Having both collagen and sequential enamel values from 10 individuals is an excellent idea. However, there is no discussion of the comparison of these two datasets. Are the pattern groups for enamel also visible in collagen? If not, what does it mean?

L. 849. The focus on wool is not that well supported by Fig. 2 and even S2. You even wrote that pointed tools on ulna are common, ubiquitous, and probably not specific to textile production. In S2, if we compare with the tools published from Rabati, or even Godedzor (Palumbi et al 2021), I see very few strong markers of wool processing in Maxta I. The two objects at the bottom of fig. 2 C have some parallels with loom weights from Rabati. Bone spindle whorls made of a sawn and drilled bovine femur head seem to be a strong marker of KA textile production. Is there any from Maxta I? If yes you should indicate their frequency and provide pictures. They would be a much better argument for wool processing than the pointed tools made of ulna.

L. 862. you should consider using the azeri spelling Çaqqallıqtəpə because there is absolutely nothing in books or internet about Chaggalligtepe

Bibliography :

There are some issues with references. I spotted a few but the rest should be checked.

References 51, 205, 206, 263 are incomplete.

46: not the right authors list.

47: incomplete author list

.

Reviewer #1: No

Reviewer #2: No

---

## [Author Response · Author response to Decision Letter 2]

20 Jan 2026

PONE-D-25-44174R1

Seasonality and mobility: An Integrative framework for reconstructing Kura-Araxes pastoral systems at Maxta I, Nakhchivan

We thank the Academic Editor and the reviewers for their detailed and constructive comments, which have greatly improved our manuscript. Below we provide a response to each point. Reviewer/Editor comments are shown in black, followed by our responses in blue.

Reviewer #1:

The second variant of the manuscript includes many changes, the outline of the paper presentation has been changed and some other changes have been incorporated. On the whole, the authors of the paper have provided explanation and clarification to most of the questions. The presented overall economic model used by one specific population group that lived at the Maxtra I settlement is based on development of pasture sheep/goat raising and cattle raising near the settlement; this population also cultivated crops. The revised text includes data of paleoarchaebotany. Generally speaking, this model is consistent with the economic model of the Caucasus inhabitants throughout the Middle Ages and the early 20th century where raising of ovicaprids played a key role.

The pending issue is to what extent raising of ovicaprids and use of adjacent pastures were correlated with the area of cultivated lands. It is clear that the author focused, mainly, on the primary role of extensive sheep/goat raising and related secondary productions; therefore, the geochemical data on the archaeozoological samples reflect a mobile nature of animal husbandry. However, the paper does not contain any multi-proxy analysis of the pasture/fodder system near the settlement and the system in the foothill area or their comparison; for this reason, the proposed interpretation remains a working hypothesis.

In the absence of the data on a significant role played by secondary products, first of all, data on the production of wool raw material, woolen cloths, and felt, I think that sheep and goats were raised to obtain meat, and, possibly, milk on a seasonal basis. I would like to note that ethnographic and modern data suggest that the lactation period of sheep in the Caucasus, probably, lasted for not more than two months or up to three months in some areas.

The issue of localization of winter pastures and fodder storage for winter has not yet been addressed. The authors analyze a potential environmental situation similar to the modern situation and think that the population could feed sheep/goats as well as cattle on pastures almost during the entire winter. The steppe landscape around Maxtra-1 seems to be consistent with this pasture strategy whereas foothill and mountainous pastures could be used during the warm period of the year. On the whole, while the authors mention that the steppe landscape and natural zone determined the type of the economic system developed by the Maxtra-I population, this statement should be spelled out with more clarity. Besides, the ethnography of the Caucasus highlanders suggests that shepherds often moved with their sheep from one area of mountainous summer pastures to another. Usually such movements lasted for 5-6 days because of overgrazing and time needed to restore pasture cover. This type of mobility should also be taken into account in developing the economic model; it is also important to take into account that, based on the ethnography of the Caucasus, only men of different ages were involved in such seasonal moves.

The proposed agro-pastoral model is based on new archaeozoological, ZooMS and isotope data, obtained for one site of the Kuro-Araxes culture and so far it remains to be the only model of this type. This model is the main outcome of the study and the basis for follow-up comparative analysis with other models that will be developed using materials from other sites. The paper is recommended for publication with minor changes and clarifications.

Thank you very much for this positive and constructive assessment. We have clarified in the revised manuscript (limitations section) that the proposed agro-pastoral model, especially the interpretation of mobility based on isotopic data, should be considered a working hypothesis in the absence of a full multi-proxy comparison of pasture and fodder systems. We also we retained a cautious interpretation of secondary products (wool/milk), given the current archaeological evidence.

Minor comments

1. Line 76 – more precise chronology of the Majkop culture of the northern Caucasus is 3650-3000 calBC. The dating of the human samples from the steppe region affected by a reservoir effect shifted the lower boundary backward to 3900 calBC, making the date older.

Thank you for alerting us to this. We have adjusted the date in text.

2. It is better to move lines 181-188 (description of Maxtra I location) up and put it before line 140 as well as descriptions of the ….. given further in the text.

Thank you for this comment. We have moved the description up.

Reviewer #2:

Thank you for the revised version of your manuscript. Although you answered to all my comments, some part of the discussion needs to be strengthened. I also have additional comments.

We are very grateful for your detailed assessment and the time you have taken to provide these helpful comments, which have significantly improved the manuscript. We have addressed each of your additional comments below.

1. L. 163 you should consider using the azeri spelling Çaqqallıqtəpə because there is absolutely nothing in books or internet about Chaggalligtepe

Thank you for this comment. We have updated the spelling throughout.

2. l. 453-454. There is a shift of approximately 5-6 month between the beginning of the crown formation and the enamel maturation. Hence the isotopic record rather reflects the life of the individual between its 6 and 18 months (see references 211 - 212 -226).

Thank you for this comment. We have added a sentence clarifying the ~5–6-month delay between crown formation onset and enamel maturation; therefore, the isotopic signal mainly reflects the individual’s life between ~6 and 18 months of age (~1 year).

“There is a delay between enamel formation and isotopic signal incorporation [210,211]. Thus, each M2 provides an isotopic record of 6 – 18 months, approximately the first year of life”

3. l. 488-499. The proposed threshold for pure C3 diet is rather low (although the upper range of the threshold in enamel is -8.58 for d13C). It is rightly stated that Janzen et al proposed a threshold at -6.7 (and not quoted Messana et al. a threshold at -7.4 in southern Europe see doi 10.1007/s12520-024-02116-z). The difference with your threshold is not discussed enough although it might highly impact the interpretation of the herd's diet.

Thank you for this comment. We acknowledge that published δ¹³C thresholds differ between studies (e.g., Janzen et al. and Messana et al.). However, thresholds derived from southern Europe are not necessarily transferable to our study area (We might be misunderstanding this part of your comment), and both Janzen et al.’s threshold and ours are inferred from the distribution of values rather than being based on local plant baseline data; therefore, neither can be considered more appropriate than the other. We are cautious about further comparing these cut-offs, as this could unintentionally imply that the Janzen threshold represents a broadly applicable reference for all Caucasus sites, which is not the case. Ultimately, the most robust approach would be to establish local baseline values for each study site/region (as e.g. done at Yeghegis-1). We have previously clarified this point in the manuscript, and we consider that the limitations are already clearly outlined in the Methods and study limitations sections. Our data will be published as open access and will be freely downloadable, so future researchers can readily use it for comparative studies, including reassessing interpretations as more robust, locally established cut-offs become available.

4. l. 552. If I understood well, 10% of the Bos primigenius bones are burnt. It means 1 bone (out of 10). On such frequency I would not comment the occurrence of burning. Same with the 9% of 11 large mammal remains. Considering that you studied approximately 25% of the assemblage, any significance of frequencies is difficult to assess.

Thank you for this helpful suggestion. We have removed the percentages from the text.

5. Fig. 4. You write Taxonomic abundance in % of the NISP but the graph shows n=410 (which includes unidentified remains). And the caprines are shown at approx 55% while in the table in supp 5 they make 68% of the NISP.

Thank you for altering us to this issue. The graph should say n=370, as it is based on the identifiable assemblage. We have therefore corrected this on Fig 4. They make up the following:

Bos 104 28.1

Sus 2 0.5

Caprines 226 61.1

Equus 9 2.4

Bos P. 10 2.7

CE 2 0.5

CC 1 0.3

Cervidae 15 4.1

Vulpes 1 0.3

Total 370 100.0

6. L. 602-608. The text is not consistent with fig. 4 and Supplementary 6. The text and fig.4 mention 20 mandibles identified with Zoom (and 5 attributed to Ovis according to the morphology). In supp6, there are only 18 specimens. None is identified as sheep in the column F (morphology). And only 13 out of 18 are identified with Zoom (column G).

Thank you for alerting us to this. The discrepancy arises because not all specimens analysed for collagen were identified using ZooMS. ZooMS was applied primarily to the subset selected for sequential enamel sampling, as well as a small number of additional specimens (some of which were used for collagen analyses and some of which were not). We also acknowledge that there was an error in the total count reported: the correct total is 18 ZooMS-analysed specimens, but these are not identical to the specimens listed in Supplementary Table 6. We have thereby updated this in text and in Fig 4. Supplementary Table 6 includes only the specimens used for sequential enamel and/or collagen analyses and therefore does not include all specimens referred to in the main text. We have clarified this in the revised manuscript to guide the reader. The full spectra and complete ZooMS results table will be made accessible via Mendeley Data (doi: 10.17632/24dyjf7c5j.1)

“We note that Table A in S6 Dataset lists only the specimens used for sequential enamel sampling and/or collagen analyses. ZooMS was performed on a wider set of mandibles, and therefore the total number of ZooMS-analysed specimens does not fully overlap with those listed in Table A. The full ZooMS results table and spectra are available on Mendeley Data doi: 10.17632/24dyjf7c5j.1.”

7. L. 619. You write "table Sx". Indeed, the average values are missing from the supplementary data. It would be interesting for the reader to have these summary statistics. Boxplots to show the distribution of d13C and d18O in the different individuals would also be helpful.

Thank you for this helpful comment. We have added the summary statistics as Table D in S6 Dataset, together with a graph.

8. L. 679-680. In variation A.2, the lowest d13C value is -10.3. You mention highland pastures. It should be discussed why -10.3 could correspond to highland pastures.

Thank you for this comment. All patterns, including A.2, are discussed in detail in the manuscript. The value of -10.3‰ simply reflects a C3 signal occurring in the summer part of the sequence, which we consider unlikely for Maxta I, where we are confident that C4 plants were available during summer. Since C4 plants have been shown to occur in patches even in winter (see Janzen and Tornero, both cited in the manuscript), it would be highly unlikely for them to be absent during the warmest season.

9. L. 703-705. In pattern 3, you assume pasture in high altitude because d13C values around -8.5 / -9 look to low for you to correspond to the Araxes valley. However, you used the same argument to identify high-altitude summer pasture in pattern 1 with d13C values around -10. It is not the same high-altitude pastures then?

In general, in the section “Sequential enamel bioapatite δ18O and δ13C values”, the values used to identify high altitude pasture are not sufficiently discussed. You don’t even mention your own article about Yeghegis-1 where values for this site at 1,500 masl are around -11 for d13C. Also, I’m not convinced by your main argument for vertical mobility when you consider that any values below the C3/C4 threshold rules out local pastures around Maxta I. The Sharur plain is one of the greenest spot of Nakhchivan, it is among the main hay provider areas of Nakhchivan. It is watered by both the Arpaçay river and the ground water of the Araxes. One can imagine green fields around the site (2-3 km radius). It is not proven that grazing on C4 plant was inevitable around Maxta I.

Thank you for this comment. We do not agree with this point. High-altitude summer pasture would be expected to be C3-dominated, and the δ¹³C of plants is also expected to vary with both altitude and season. In contrast, Maxta I in summer represents a mixed C3/C4 environment, as demonstrated by multiple studies (cited in the manuscript). Therefore, the herders may have moved to pastures at different altitudes, or the variation could simply reflect seasonal changes in the local environment. You also mention the Sharur plain as one of the greenest areas, suggesting that green fields may have existed within a 2–3 km radius of the site, and thus that grazing on C4 plants was not necessarily inevitable around Maxta I. However, unless the Sharur plain is at an altitude too high for C4 plants, it is not clear why hay would not also include some C4 contribution, as these plants are seasonal and would be expected to grow during the warmest part of the year.

Concerning the environmental setting of the Sharur plain, we acknowledge that local conditions are heterogeneous. However, characterising it as a purely C3 environment would not be accurate. Present-day climatic data indicate an annual precipitation of approximately 300 mm, which classifies the area as semi-arid. Such conditions support a mixed C3/C4 biome, particularly during the drier months. C4 grasslands have been documented in comparable settings in the bordering area of northwestern Iran (Hatami and Khosravi, 2013). In these contexts, C4 Chenopodiaceae species are known to persist until late autumn, and some may survive even into winter in protected microhabitats. Given this evidence, it is highly unlikely for Maxta I to be situated in a purely C3 environment year-round.

Furthermore, both Tornero et al. (2016) and Janzen (2023), based on ancient and modern faunal assemblages, indicate that animals grazing in lowland environments may display a more C4-enriched signal during the winter months (when C4 plants persist in dry or sheltered microhabitats), and a more C3-influenced signal during the summer only when herds access higher-elevation pastures where C3 vegetation dominates. Tornero et al. (2016) also describe these lowlands as mixed C3/C4 environments in summer. Overall, an environment that supports C4 plants in winter is unlikely to lack them entirely during summer, given that these plants typically thrive under warmer conditions.

Yeghegis-1 has already been cited in the text, together with other relevant studies from the Caucasus. However, we note that directly comparing absolute δ¹³C and δ¹⁸O values across comparative datasets is not straightforward, given differences in chronology (spanning millennia), environmental context, and potential methodological variation (e.g., sampling strategy and preparation protocols). We therefore prefer to focus on the internal patterns within our dataset and interpret them within the local context of Maxta I.

10. L. 710 About the section “seasonality of birth”. I checked the raw data you provided in S6 and S7. I think that it would be necessary to provide the pictures of the sampled teeth, so that the reader can see how the sampling was made and can see the ERJ.

For example, I have a question about the distance to erj of the samples of MI 18. There should be a gap of at least 1 mm between two samples. Because the procedure is to measure the sample position on the upper ridge (occl

---

## [Decision Letter · Decision Letter 2]

15 Mar 2026

Seasonality and mobility: An Integrative framework for reconstructing Kura-Araxes pastoral systems at Maxta I, Nakhchivan

PONE-D-25-44174R2

Dear Dr. Maurer,

We’re pleased to inform you that your manuscript has been judged scientifically suitable for publication and will be formally accepted for publication once it meets all outstanding technical requirements.

Kind regards,

Vanessa Carels

Staff Editor

PLOS One

Additional Editor Comments (optional):

Reviewers' comments:

Reviewer's Responses to Questions

**Comments to the Author**

Reviewer #1: All comments have been addressed

Reviewer #2: All comments have been addressed

2. Is the manuscript technically sound, and do the data support the conclusions?

Reviewer #1: Yes

Reviewer #2: Yes

3. Has the statistical analysis been performed appropriately and rigorously?

Reviewer #1: Yes

Reviewer #2: Yes

4. Have the authors made all data underlying the findings in their manuscript fully available?

Reviewer #1: Yes

Reviewer #2: Yes

5. Is the manuscript presented in an intelligible fashion and written in standard English?

Reviewer #1: Yes

Reviewer #2: Yes

Reviewer #1: The third variant of the manuscript includes important additional information and clarifications, enabling the scholars to discuss major components of economic strategy at Maxtra I, which is a Kura-Araxes culture site in the southern Caucasus. I think that the main task of the study has been addressed, the paper provides new data on zooarchaeology, ZooMS and stable isotope analyses as well as additional paleobotanical and settlement data. Despite the limitation of the samples analyzed and a lack of some very important environmental data, the authors described in detail the role of domesticated animals, especially sheep/goats, in the economic model developed by the Maxtra I population and human-animal relationship. They show that the agropastoral system led to development of the system of seasonal pastures and seasonal movement of a greater part of the herd from the settlement to highland pastures. The animals that were kept at the site grazed in the nearby pastures as evidenced by the isotope values in the animal bones.

The model described shows a diversity of agropastoral economy based on the adaption of animal husbandry to seasonal changes in the local environment and a need to meet everyday demand of the population in food, secondary products and other resources.

This version can be recommended for publication in Plos One. Minor comment: the authors must check if all their references in the text are in square brackets.

Reviewer #2: I appreciate the authors for their thorough responses to all my earlier comments. The data has been verified, the figures have been enhanced, and the supplementary materials now include all the essential raw data.

However, the authors and I continue to have significant disagreements regarding the classification of highland and lowland pastures based on the d13C values. Nevertheless, the authors have presented all their reasoning for their hypothesis in the text, with appropriate cautions, and the raw data is accessible for future reinterpretation. There is no further need to debate this interpretation.

For these reasons, I recommend the publication of this manuscript.

.

Reviewer #1: No

Reviewer #2: No

---

## [Editor Report · Acceptance letter]

PONE-D-25-44174R2

PLOS One

Dear Dr. Maurer,

I'm pleased to inform you that your manuscript has been deemed suitable for publication in PLOS One. Congratulations! Your manuscript is now being handed over to our production team.

Kind regards,

on behalf of

Dr. Vanessa Carels

Staff Editor

PLOS One